neuroscience/biomechanics

human gait, interlimb coordination, generalization, motor adaptation, non-motorized treadmill

**Author for correspondence:**
Julia T. Choi
e-mail: juliachoi@ufl.edu

# Contributions of spatial and temporal control of step length symmetry in the transfer of locomotor adaptation from a motorized to a non-motorized split-belt treadmill

Daniel L. Gregory[1], Frank C. Sup IV[2] and Julia T. Choi[1,3,4]

[1]Department of Kinesiology, [2]Department of Mechanical and Industrial Engineering, and [3]Neuroscience and Behavior Program, University of Massachusetts Amherst, Amherst, MA, USA
[4]Department of Applied Physiology and Kinesiology, University of Florida, PO Box 118205, Gainesville, FL 32611-8205, USA

(iD) JTC, 0000-0003-4963-9094

Walking requires control of where and when to step for stable interlimb coordination. Motorized split-belt treadmills which constrain each leg to move at different speeds lead to adaptive changes to limb coordination that result in after-effects (e.g. gait asymmetry) on return to normal treadmill walking. These after-effects indicate an underlying neural adaptation. Here, we assessed the transfer of motorized split-belt treadmill adaptations with a custom non-motorized split-belt treadmill where each belt can be self-propelled at different speeds. Transfer was indicated by the presence of after-effects in step length, foot placement and step timing differences. Ten healthy participants adapted on a motorized split-belt treadmill (2 : 1 speed ratio) and were then assessed for after-effects during subsequent non-motorized treadmill and motorized tied-belt treadmill walking. We found that after-effects in step length difference during transfer to non-motorized split-belt walking were primarily associated with step time differences. Conversely, residual after-effects during motorized tied-belt walking following transfer were associated with foot placement differences. Our data demonstrate decoupling of adapted spatial and temporal locomotor control during

transfer to a novel context, suggesting that foot placement and step timing control can be independently modulated during walking.

# 1. Introduction

A healthy human nervous system can rapidly adapt how the limbs move to meet changing environmental demands. Walking on a motorized split-belt treadmill initially induces asymmetric step lengths when the two limbs are constrained to move at different speeds [1,2]. Adaptation to restore symmetric step lengths via error-based learning mechanisms during split-belt treadmill walking involves changes in both spatial control of foot placement and temporal control of step timing [3–5]. These spatial and temporal motor outputs reflect distinct neural strategies to overcome the asymmetric belt speeds imposed during motorized split-belt walking [4,6,7]. The newly adapted foot placement and step timing are stored post-adaptation, resulting in after-effects represented by a step length asymmetry in the opposite direction on return to the tied-belts condition when both belts move at the same speed [8].

Partial or complete transfer of locomotor adaptation to an untrained walking condition (e.g. motorized treadmill to over-ground walking), as measured by the magnitude of after-effects, is thought to reflect the adaptation of overlapping or shared neural circuits [9–12]. Conversely, lack of adaptation transfer (e.g. between forward and backward walking) is thought to reflect adaptation of independent neural circuits [13]. Changes in walking condition, such as the speed of walking, can lead to diminishing after-effects as the speed deviates from that of the slow limb during adaptation [12]. Further, after-effect magnitude during transfer is known to be modulated by contextual clues such as when transferring to over-ground walking, or when vision is manipulated between training and transfer contexts [11,14]. The removal of vision during training and transfer maximizes transfer in healthy adults, but when vision is only removed during training or transfer conditions limits the transfer of spatial (i.e. step length) or temporal (i.e. phase shift) parameters, respectively [11]. These studies have demonstrated that the transfer of locomotor adaptations is consequent to different constraints of walking. However, it is unclear how the expression of after-effects may be influenced when the limbs have greater freedom to express adapted limb coordination.

Here, we developed a non-motorized, split-belt treadmill that allows the user to set their own walking pace while simultaneously allowing for asymmetric behaviour. In contrast with motorized treadmills, non-motorized treadmills have freely movable belts which are driven by participants pushing against an inclined or concave surface which allow participants to self-select and express natural gait variability [15–17]. During non-motorized split-belt treadmill walking, the speed of each belt/limb is user controlled. This novel device thus allows us to examine the adaptation and storage of spatial and temporal motor outputs during walking in the absence of leg-specific speed constraints. Previous research indicated that after-effects from split-belt walking adaptation are modulated by speed constraints, with the largest after-effects occurring while both legs move at the slow speed [12]. Using the non-motorized split-belt treadmill, we can assess whether self-selected walking speeds are also modulated after motorized split-belt walking adaptation, and how self-selected limb speeds influence after-effect magnitude.

In this study, we examined the transfer of adaptation from motorized split-belt treadmill walking at a 2 : 1 speed ratio to non-motorized split-belt walking at self-selected speeds on each side. We predicted three potential outcomes. (i) Partial transfer and full washout. We hypothesized that in the absence of asymmetrical changes in self-selected leg speeds during non-motorized split-belt walking, the transfer of adapted foot placement (spatial control) and step timing (temporal control) from motorized to non-motorized split-belt walking would result in asymmetrical step length after-effects typical of motorized split-belt adaptation and would lead to the washout of the adapted motor pattern [8]. (ii) Full transfer and no washout. Alternatively, we hypothesized that self-selected speed would increase on the leg that was on the fast belt and decrease on the leg that was on the slow belt to match speeds during adaptation. Asymmetric changes in leg speed that complement the adapted foot placement and step time would result in symmetrical step lengths during non-motorized split-belt walking such that the adapted spatio-temporal walking pattern during motorized split-belt walking at 2 : 1 speed ratio would persist during transfer to non-motorized split-belt walking and not washout. (iii) Partial transfer and partial washout. In addition, the adapted foot placement and step timing contributions to step length difference may show only partial (incomplete) transfer to non-motorized

split-belt walking. Partial transfer and washout of after-effects would result in residual after-effects during subsequent tied-belt motorized split-belt walking.

# 2. Methods

## 2.1. Participants

Ten healthy volunteers (six female, four male; age 26.5 ± 5.6 years) with no neurological or biomechanical impairments were recruited for this study. All study protocols were approved by the University of Massachusetts, Amherst Institutional Review Board. All participants provided written informed consent prior to enrolment. None of the participants had prior experience walking on a split-belt treadmill.

## 2.2. Experimental set-up

### 2.2.1. Motorized split-belt treadmill

Participants walked on a split-belt treadmill (Bertec Corp., Columbus, OH, USA) that has separate left and right belts, each with its own motor. During motorized treadmill walking, participants wore a non-weight-bearing safety harness suspended from the ceiling. Participants were instructed to minimize handrail use, walking with normal arm swing, and to maintain forward gaze. Participants were randomly assigned left or right limb to the slow belt during split-belt walking and subsequent references to a limb will be as slow or fast regardless of condition.

### 2.2.2. Non-motorized spit-belt treadmill

We designed and built a user-propelled non-motorized split-belt treadmill (figure 1a). The non-motorized treadmill was fabricated from two commercially available non-motorized treadmills (Inmotion II Manual Treadmill; Stamina, Springfield, MO, USA), which were designed to share a common support structure and minimize spacing between the belts (32 mm). To minimize friction with the belt-deck interface, a sheet of polytetrafluoroethylene (PTFE, 0.30′ sheet; ePlastics, San Diego, CA, USA) plastic was secured to each treadmill deck. An approximately 1 kg mass was added to each flywheel to increase the inertial properties of the belt-flywheel system to ensure continuous movement as the limb transitioned from stance to swing, enabling smoother transition into the next stance phase. The treadmill uses gravity (approx. 13° incline) and the users body weight to assist driving the symmetrically resisted but independently user-propelled belts. During non-motorized treadmill walking, participants lightly held the handrails and were instructed to avoid supporting body weight.

## 2.3. Split-belt adaptation paradigm

The adaptation paradigm (figure 1b) consisted of motorized tied-belt, motorized split-belt and non-motorized split-belt walking conditions. 'M' will be used to indicated motorized treadmill conditions and 'N' non-motorized treadmill conditions. Participants walked on the motorized and non-motorized treadmills for 5 min each at the medium speed (1.0 m s$^{-1}$) and at preferred speed, respectively, to familiarize with walking on the different treadmills. Following familiarization, participants were recorded walking on the motorized treadmill with belts tied at a slow (0.67 m s$^{-1}$), medium (1.0 m s$^{-1}$), and fast (1.34 m s$^{-1}$) speed in randomized order to minimize any potential influence of motorized treadmill walking speed on subsequent non-motorized treadmill walking speed. The non-motorized split-belt treadmill baseline was recorded at preferred walking speed. Each baseline trial lasted 2 min. Participants were instructed to 'walk as fast as you would to a meeting for which you have adequate time to arrive'. During *Adaptation-M*, participants walked on the motorized split-belt at a 2 : 1 speed ratio (0.67 and 1.34 m s$^{-1}$) for 10 min. The split-belt condition was introduced with both belts accelerating from zero at 1.0 m s$^{-2}$. These speeds were chosen because the average speed is 1.0 m s$^{-1}$, which was the average preferred walking speed on the non-motorized treadmill during pilot testing. Participants were then instructed to side-step from the motorized treadmill onto the non-motorized treadmill less than 2 feet away. Forward stepping was avoided to prevent washout of after-effects. During *Transfer-N*, participants walked for 5 min on the non-motorized treadmill. Participants returned to the motorized treadmill for the *Washout-M* period and walked at the slow speed for an additional 5 min. The slow speed was used during washout because this is the speed with which after-effects have

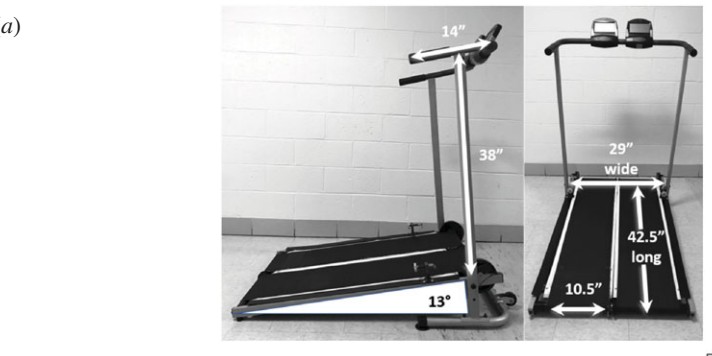

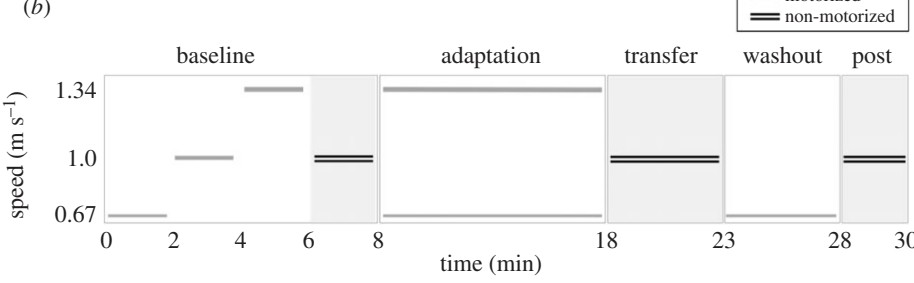

**Figure 1.** Non-motorized split-belt treadmill and paradigm. (*a*) Non-motorized treadmill dimensions (inches), (left panel) handrail height from top of treadmill deck and handle depth, and deck angle in degrees. Non-motorized treadmill dimensions, (right panel) deck width and length and individual belt widths. (*b*) Split-belt treadmill transfer paradigm. Grey lines indicate motorized treadmill walking while double black lines in grey shading indicate non-motorized treadmill walking at preferred speeds.

been demonstrated to be the largest [12]. During *post-Washout-N*, participants again walked on the non-motorized treadmill for 2 min.

## 2.4. Data collection

During motorized treadmill walking, kinematics were recorded at 100 Hz using a four-camera motion capture system (Qualisys, Sweden). Reflective markers were placed bilaterally over the fifth metatarsal, lateral malleolus, tibial plateau, greater trochanter and the anterior superior iliac spines. During non-motorized treadmill walking, kinematics were relegated to the use of only ankle markers for the identification of spatio-temporal parameters due to limitations in capture space during data collection.

## 2.5. Data analysis

Data analysis was performed in Matlab (Mathworks, Natick, MA, USA; v. R2017a). Marker data were low-pass filtered at 6 Hz with a second-order Butterworth filter. For both motorized and non-motorized treadmill walking, we defined heel contact and toe-off as the time of peak anterior and posterior ankle position for each step, respectively. Note that during motorized treadmill conditions, fast and slow speeds are fixed, whereas during non-motorized treadmill walking, the speed of each belt/limb is user controlled and becomes a third degree of freedom while walking; stance limb velocity during walking on the non-motorized treadmill was calculated as the instantaneous speed of the ankle marker at the time of contralateral heel strike.

The main outcome measure was step length difference, which has been repeatedly shown to characterize adaptation and learning processes in locomotion [1,18]. Step length is defined as the anterior–posterior distance between the ankle markers at the time of heel contact (figure 2*a*) [1]. Step time is defined as the time interval between successive heel contacts, where the slow step time ($t_s$) is the duration between heel contact on the fast limb to the subsequent heel contact on the slow limb and vice versa for the fast step time ($t_f$). To capture the independent spatial and temporal contributions to step length difference, we calculated step lengths and step length difference analytically [3]. Slow analytical step length is calculated as

$$\mathrm{SL}_s = \alpha_s + v_f \cdot t_f \tag{2.1}$$

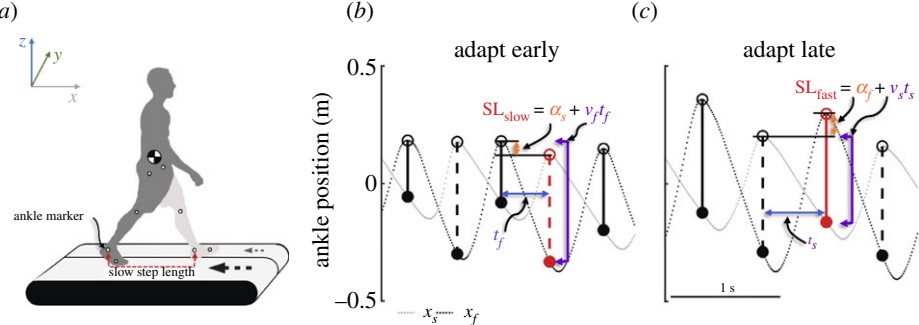

**Figure 2.** Spatial and temporal step length components. (*a*) Walker demonstrating the position of the reflective markers on the fifth metatarsal, lateral malleolus and lateral femoral condyle. Black and white checkered circle represents the estimated centre of mass used to calculate the reference frame for ankle marker data on the motorized treadmill. Slow step length shown (red arrow line) as distance between ankle markers at the time of slow limb heel contact. Grey and black arrows indicate belt speeds. (*b,c*) Ankle marker trajectories for the fast (black, $x_f$), and slow (grey, $x_s$) limbs in the centre of mass reference frame. Positive values indicate positions in front of the centre of mass. Open circles indicate the position and time of the leading limb at heel contact. Filled circles indicate the position and time of the trailing limb at the time of leading limb heel contact. Solid vertical lines connecting open and closed circles indicate the instantaneous fast step length and the dashed lines connecting open and closed circles indicate the instantaneous slow step length. (*b*) The spatial component of the slow step length ($SL_{slow}$) is determined by the relative position of the leading leg at heel contact on the slow leg ($\alpha_s$). The temporal component of the slow step length is determined by the velocity of the fast leg multiplied by fast step time ($v_f * t_f$). (*c*) The fast step length ($SL_{fast}$) can be estimated by the summation of the spatial component ($\alpha_f$), and the product of the velocity of the slow limb ($v_s$) and the slow step time ($t_s$).

where the first term ($\alpha_s$) is the sum of the ankle position of the slow leg at slow heel contact relative to the ankle position of the fast leg at fast heel contact (figure 2*b* orange), and the product of the velocity of the fast limb ($v_f$) and the fast step time ($t_f$, blue). The $v_f t_f$ component is the distance travelled by the fast limb from fast heel strike to fast position at slow heel strike (purple). The fast step length is calculated in the same way (figure 2*c*). Step length difference ($SL_{diff}$) was defined as $SL_{fast} - SL_{slow}$, where $SL_{fast}$ is the step length with fast leg leading and $SL_{slow}$ is with the slow leg leading at heel strike. To quantify the spatial, temporal and velocity contribution to changes in step length difference ($SL_{diff}$), we applied the following analytical model of step length difference (for derivation, see Finley *et al.* [3])

$$SL_{diff} = \Delta\alpha + \bar{v} \cdot \Delta t + \bar{t} \cdot \Delta v, \tag{2.2}$$

where the first term (spatial component) represents the difference between the relative positions of the feet at the fast and slow heel strikes ($\Delta\alpha$), the second term (temporal contribution) is the difference in slow and fast step times ($\Delta t$) as a function of average foot speed ($\bar{v}$), and the third term (velocity component) is the difference in slow and fast foot velocities ($\Delta v$) as a function of the average step time ($\bar{t}$) (figure 2*b,c*). $\Delta\alpha$ is positive when the foot position of the fast leg at fast heel contact is anterior to the foot position of the slow leg at slow heel contact. $\Delta t$ is positive when the duration from slow heel contact to fast heel contact ($t_s$) is longer than the duration from fast heel contact to slow heel contact ($t_f$). Electronic supplementary material, figure S4*a* shows the accuracy of the analytical step length calculation for estimating the instantaneous step length across slow, medium, fast and adapted walking conditions. Note that for the step length analysis during motorized treadmill walking, we estimated the centre of mass position by taking the average position of the two hip markers; the anterior–posterior centre of mass position was then subtracted from each ankle marker to express foot position in the centre of mass reference frame (figure 2*a*). Since hip markers could not be recorded during non-motorized treadmill walking, the average value of the ankle marker across a trial from the limb assigned to the slow belt during motorized treadmill walking was subtracted from each ankle marker's anterior–posterior position to obtain a 'body-centred' reference frame. Positive values indicate foot positions in front of the centre of mass and negative indicate positions behind the centre of mass.

Finally, we calculated double support time as the difference between stance time and step time [5]. Double support difference was calculated as the difference between fast and slow double support times. For a derivation of the analytical double support time, see the electronic supplementary material.

## 2.6. Statistical analysis

Group means were calculated in epochs: across the last 10 strides for each baseline trial (motorized slow, motorized medium, motorized fast and non-motorized preferred), and the first five strides (early) and last five strides (late) for *Adaptation-M*, *Transfer-N*, *Washout-M* and *post-washout-N* trials. Student's *t*-tests were used to compare baseline values with zero to determine whether baseline asymmetries were present. Separate repeated-measures ANOVAs were used to test the effects of epoch on step length difference and its components. *Post hoc* analyses with the Bonferroni corrections were used to assess for differences between each epoch compared with the corresponding baseline (e.g. baseline-M versus early adapt, baseline-M versus late adapt), and between early and late epochs (e.g. early adapt versus late adapt) when significant main effects were determined. A repeated-measures ANOVA was used to test the effects of epoch (baseline-N, *early Transfer-N*, *late Transfer-N* and *post-Washout-N*) and limb on belt speed during non-motorized treadmill walking. To investigate whether there was any partial de-adaptation during *Transfer-N* that influenced *Washout-M* after-effects, we performed a linear regression of the after-effects between *early Transfer-N* and *early Washout-M* periods. We also performed a repeated-measures ANOVA to test the effects of epoch (*Transfer-N early*, *Washout-M early*) and limb on individual step lengths as a comparison of previous split-belt treadmill research involving the use of an incline. All statistics were performed using JASP (v. 0.13.1.0) with $\alpha$ levels set to $p = 0.05$.

# 3. Results

## 3.1. Step length adaptation

Repeated-measures ANOVA revealed significant effects of epoch on step length difference ($F_{8,72} = 24.7$, $p < 0.001$), and the components of step velocity ($F_{3.23,29.1} = 75.6$, $p < 0.001$), step time ($F_{8,72} = 16.5$, $p < 0.001$) and step position ($F_{3.56,32.0} = 27.1$, $p < 0.001$). Group-averaged step length difference and its components were not significantly different from zero during baseline motorized and non-motorized treadmill walking trials (all $p > 0.05$). Figure 3a shows gradual changes in step length difference (dark grey/black) during the first 100 and last 20 strides in adaptation. Note that during the *early Adaptation* period, step length difference shows a gradual increase in asymmetry from stride 1 to about stride 10. Therefore, our planned analysis using the first five strides during *early Adaptation* is an underestimate of the maximal asymmetry achieved during split-belt treadmill walking. During *early Adaptation-M*, step length difference showed a large initial change compared with *baseline-M* ($t = 5.7$, $p < 0.001$, figure 3b). By *late Adaptation-M*, step length difference was reduced back to baseline levels ($t = 0.5$, $p = 1.0$, figure 3b).

Step length difference was decomposed into independent step velocity (magenta), step position (spatial, red) and step time (temporal, blue) contributions (figure 3c,d). The step velocity component was significantly different from slow *baseline-M* in *early Adaptation-M* ($t = 11.5$, $p < 0.001$), and became increasingly negative from early to late adaptation, reflecting a large negative velocity-induced perturbation which required opposition ($t = 4.0$, $p = 0.005$, figure 3d). During *early Adaptation-M*, the step position component showed a significant initial asymmetry relative to baseline ($t = -4.8$, $p < 0.001$, figure 3d). The step time component showed a more gradual change, with *early Adaptation-M* values not significantly different from baseline-M ($t = -1.2$, $p = 1.0$, figure 3d). By late *Adaptation-M*, the step position ($t = -10.7$, $p < 0.001$) and step time ($t = -8.8$, $p < 0.001$) components both increased significantly relative to baseline to cancel the negative step velocity component, resulting in symmetric step lengths that are not different from baseline ($t = 0.51$, $p = 1.0$, figure 3d).

## 3.2. Step length transfer

We found significant transfer of motorized treadmill adaptations to the non-motorized split-belt treadmill. Figure 4a shows the after-effect in step length difference during the first 100 and last 20 strides in *Transfer-N*. During *early Transfer-N*, step length difference showed a significant after-effect relative to *baseline-N* ($t = -5.0$, $p < 0.001$, figure 4b). By *late Transfer-N*, step length difference returned to baseline levels ($t = -0.67$, $p = 1.0$) and was significantly different from *early Transfer-N* ($t = 4.3$, $p = 0.002$).

Figure 4c,d shows the contribution of the step velocity, position and step timing components to the step length difference during *Transfer-N*. The step velocity component was not significantly different from *baseline-N* in *early Transfer-N* ($t = -2.8$, $p = 0.3$, figure 4d) and *late Transfer-N* ($t = 1.5$, $p = 1.0$). The step position component was not significantly different from *baseline-N* during *early Transfer-N* ($t = -0.58$,

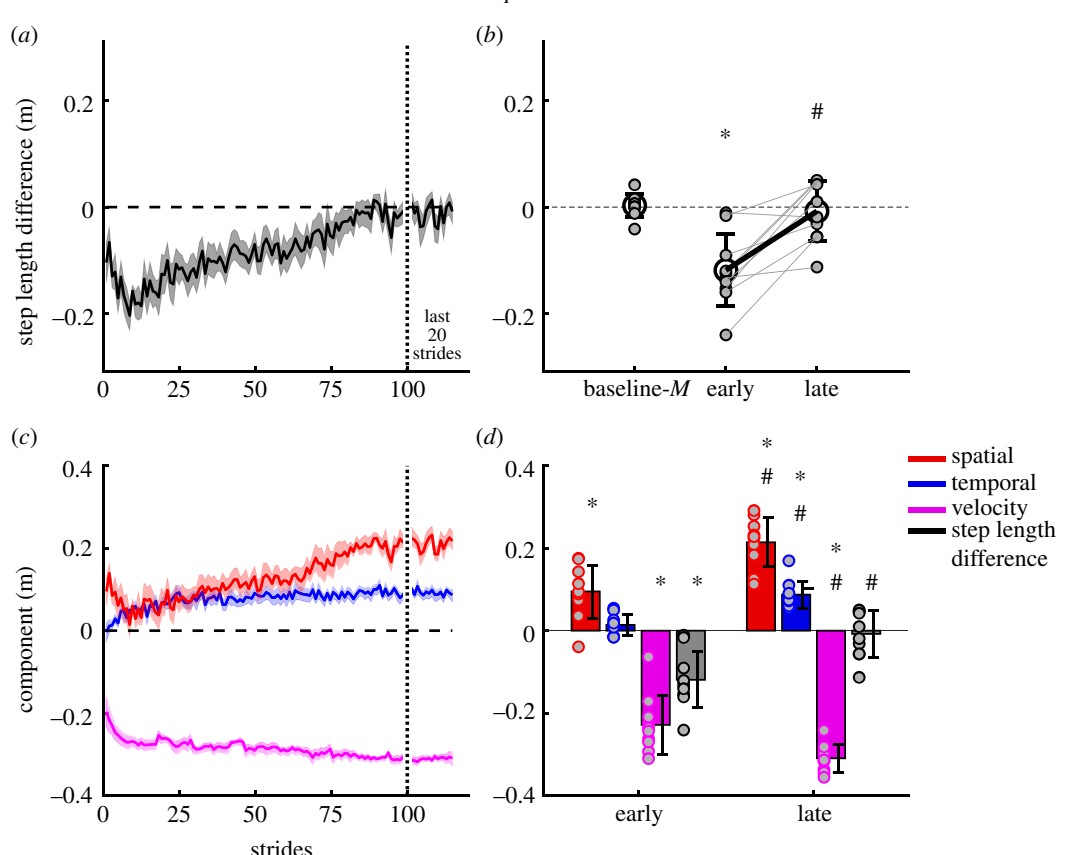

**Figure 3.** *Adaptation-M*; progression of step length difference and contribution of spatial, temporal and velocity asymmetries to motorized split-belt treadmill walking during adaptation. (a) Group average step length difference during the first 100 and last 20 strides of the adaptation period. Negative values indicated the slow step length is greater than the fast step length. (b) Group mean (large black circles, ±s.d.) and individual subject step length difference data (small grey filled circles) comparing baseline-M slow walking with early adapt and late adapt. Light grey lines connect individual subject data points between adapt early and adapt late. (c) Group mean stride-by-stride changes in individual components for the first 100 and last 20 strides; spatial (red), temporal (blue) and velocity (magenta) differences. (d) Group mean (±s.d.) and individual subject component data (grey filled coloured circles) comparing adapt early and adapt late. Shaded area represents ±s.e. *Significant difference to baseline slow; #late epoch significantly different from early epoch.

$p = 1.0$, figure 4d), and there were no significant differences between *early* and *late Transfer-N* for the step position component ($t = -1.5$, $p = 1.0$). The step position component was also not different from baseline at *late Transfer-N* ($t = -2.0$, $p = 1.0$, figure 4d). These results indicate that the spatial component did not contribute to the initial after-effects in step length difference. The step time component, however, was significantly different from *baseline-N* during *early Transfer-N* ($t = -4.0$, $p = 0.005$, figure 4d), being the primary contributor to the large step length differences. By *late Transfer-N*, the step time component decreased back to baseline values ($t = -0.26$, $p = 1.0$), indicating washout of the temporal after-effect.

## 3.3. Step length washout

After-effects in step length difference reappeared during motorized treadmill walking in *early Washout-M*. Step length difference was significantly different from *baseline-M* during *early Washout-M* ($t = -6.7$, $p < 0.001$, figure 5a) and returned back to baseline by *late Washout-M* indicated by significant differences from *early* to *late Washout-M* ($t = 5.4$, $p < 0.001$) and no differences with baseline values ($t = -1.3$, $p = 1.0$, figure 5b).

Figure 5c shows the contribution of the step velocity, step position and step timing components to the step length difference during the *Washout-M* period. The step velocity component was not different from *baseline-M* during *early Washout-M* ($t = 0.41$, $p = 1.0$) or *late Washout-M* ($t = -0.016$, $p = 1.0$), due to the symmetric belt speeds (figure 5c,d). The step position component displayed significant after-effects

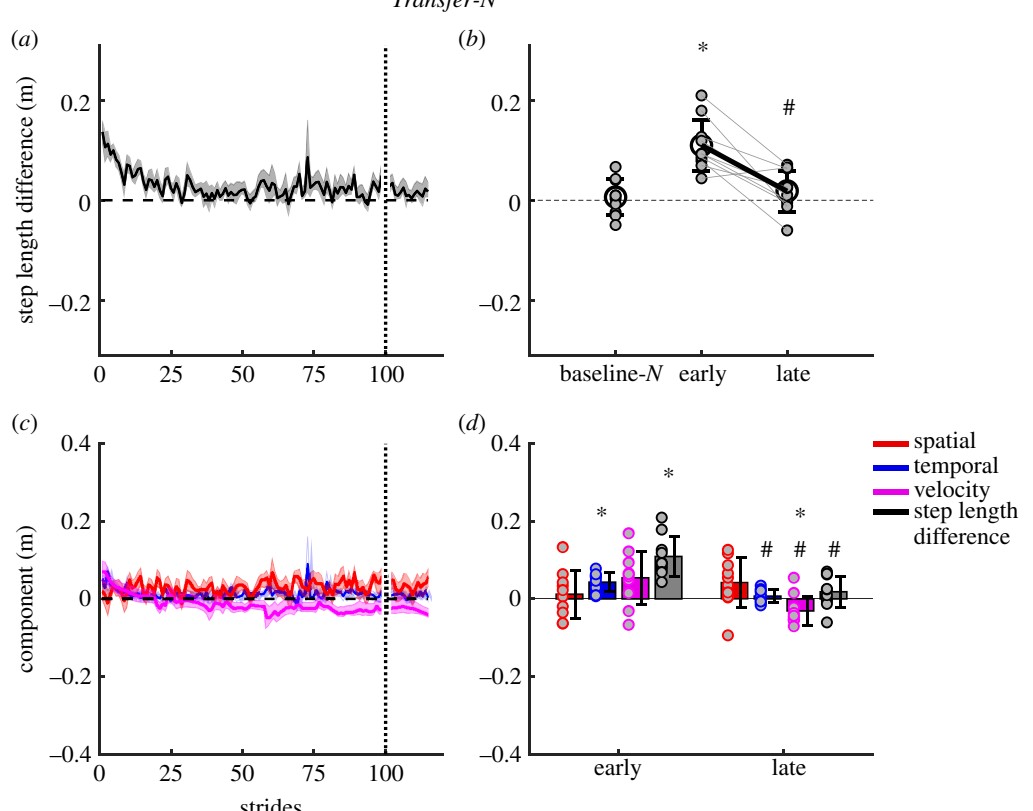

**Figure 4.** *Transfer-N*; progression of step length difference and contribution of spatial, temporal and velocity asymmetries to non-motorized split-belt treadmill walking during transfer. (*a*) Group average step length difference during the first 100 and last 20 strides of the transfer period. (*b*) Group mean (±s.d.) and individual subject step length difference data (small grey filled circles) comparing baseline-N walking with *early Transfer-N* and *late Transfer-N*. (*c*) Group mean stride-by-stride changes in individual components. (*d*) Group mean (±s.d.) and individual subject component data (grey filled coloured circles) comparing transfer early and transfer late. Shaded area represents ±s.e. \*Significant difference to baseline-N; #late epoch significantly different from early epoch.

during *early Washout-M*, ($t = -6.2$, $p < 0.001$) that reduced back to baseline by *late Washout-M* ($t = -1.0$, $p = 1.0$, figure 5*d*). The step time component was not significantly different from *baseline-M* during *early Washout-M* ($t = -2.56$, $p = 0.4$) nor *late Washout-M* ($t = -0.59$, $p = 1.0$, figure 5*d*).

## 3.4. Comparisons between early transfer and early washout

Because split-belt adaptation on an incline has been shown previously to impact locomotor adaptation and after-effects via differences in slow and fast step lengths [8], we also tested the step lengths of individual limbs during the *early Transfer-N* and *early Washout-M* periods. Repeated-measures ANOVA revealed no significant effects of epoch ($F_{1,5} = 0.151$, $p = 0.707$, $\eta^2 = 0.004$) or epoch × limb interaction ($F_{1,5} = 3.624$, $p = 0.707$, $\eta^2 = 0.004$), but significant effects of limb ($F_{1,5} = 102.894$, $p < 0.001$, $\eta^2 = 0.63$) on individual step lengths (figure 6*a*).

Linear regression analysis showed that step length difference during *early Transfer-N* did not predict the step length difference during *early Washout-M* ($p = 0.782$, figure 6*b*), suggesting that transfer did not result in unlearning of motorized treadmill adaptation. Linear regression analysis on the individual components also showed no interaction between *early Transfer-N* and *early Washout-M* for the step position ($p = 0.132$), step time ($p = 0.451$) or velocity components ($p = 0.457$). Additionally, there was no difference between the magnitude of *early Transfer-N* and *early Washout-M* after-effects ($t = -1.544$, $p = 1.0$).

## 3.5. Non-motorized walking speeds

Because the non-motorized treadmill has a third degree of freedom (i.e. limb speed), we assessed the effects of adapting to motorized treadmill split-belt walking to non-motorized treadmill limb

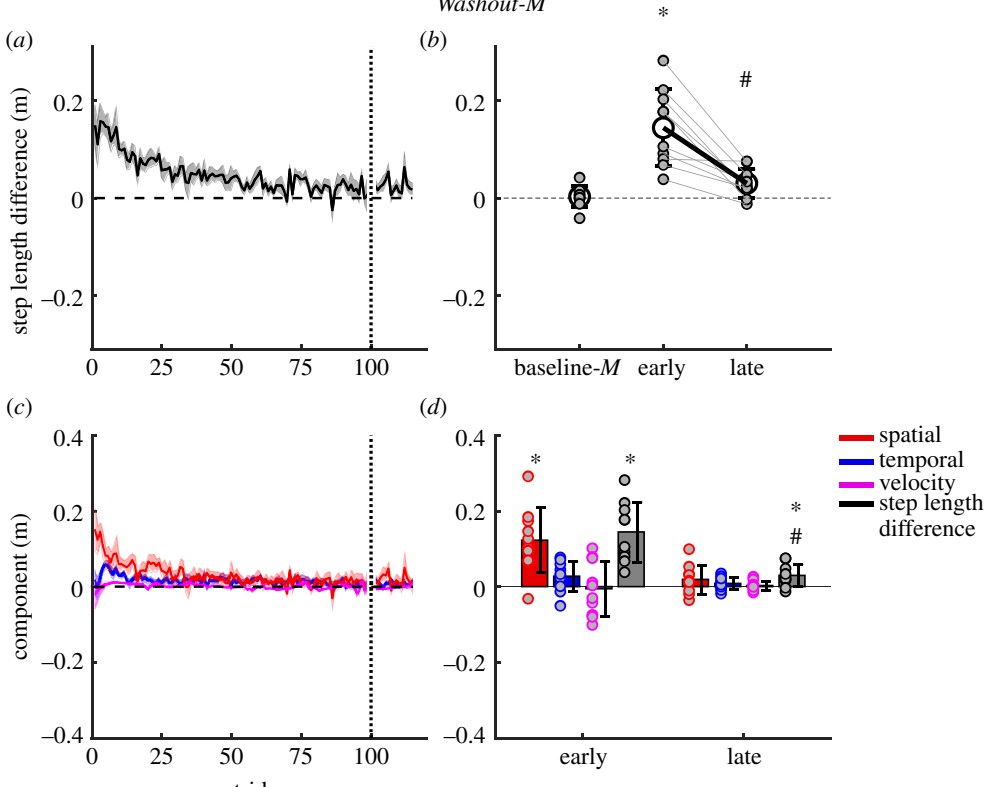

**Figure 5.** *Washout-M*; progression of step length difference and contribution of spatial, temporal and velocity asymmetries to motorized treadmill walking during washout. (*a*) Group average step length difference during the first 100 and last 20 strides of the adaptation period. (*b*) Group mean (±s.d.) and individual subject step length difference data (small grey filled circles) comparing baseline-*M* slow walking with early washout and late washout. (*c*) Group mean stride-by-stride changes in individual components for the first 100 and last 20 strides. (*d*) Group mean (±s.d.) and individual subject component data (grey filled coloured circles) comparing washout early and washout late. Shaded area represents ±s.e. *Significant difference to baseline slow, #late epoch significantly different from early epoch.

speed. We found significant effects of epoch ($F_{3,27} = 15.1$, $p < 0.001$, $\eta^2 = 0.435$) and an interaction of epoch × limb ($F_{3,27} = 8.11$, $p = 0.006$, $\eta^2 = 0.096$). *Post hoc* analysis showed that *early Transfer-N* was significantly different from baseline ($t = 5.653$, $p < 0.001$, figure 6*c*) and *late Transfer-N* ($t = -4.209$, $p = 0.002$). *Post-washout-N* walking limb speed was significantly slower than baseline ($t = 5.054$, $p < 0.001$) and *late Transfer-N* ($t = 3.61$, $p = 0.007$). The fast limb was significantly slower than the slow limb during *early Transfer-N* ($t = -3.653$, $p = 0.030$), but there were no differences between limbs at *late Transfer-N* ($t = 1.959$, $p = 1.0$). There were no differences between individual limb speeds during baseline, or *post-Washout-N* walking on the non-motorized treadmill.

## 3.6. Step length post-washout

No residual step length difference after-effects were present once returned to the non-motorized treadmill *post-Washout-N* ($t = 0.108$, $p = 1.0$, not shown). In addition, there were no asymmetries in any individual component during *early post-Washout-N* walking (all $p > 0.05$).

## 3.7. Double support

Repeated-measures ANOVA revealed significant effects of epoch on double support difference ($F_{3.2,28.6} = 53.1$, $p < 0.001$, figure 7). Group-averaged double support difference was not significantly different from zero during motorized and non-motorized split-belt treadmill walking (all $p > 0.05$). Figure 7*a* shows gradual changes in double support difference during the first 100 and last 20 strides in *Adaptation-M*. During *early Adaptation-M*, double support difference showed a large initial change compared with

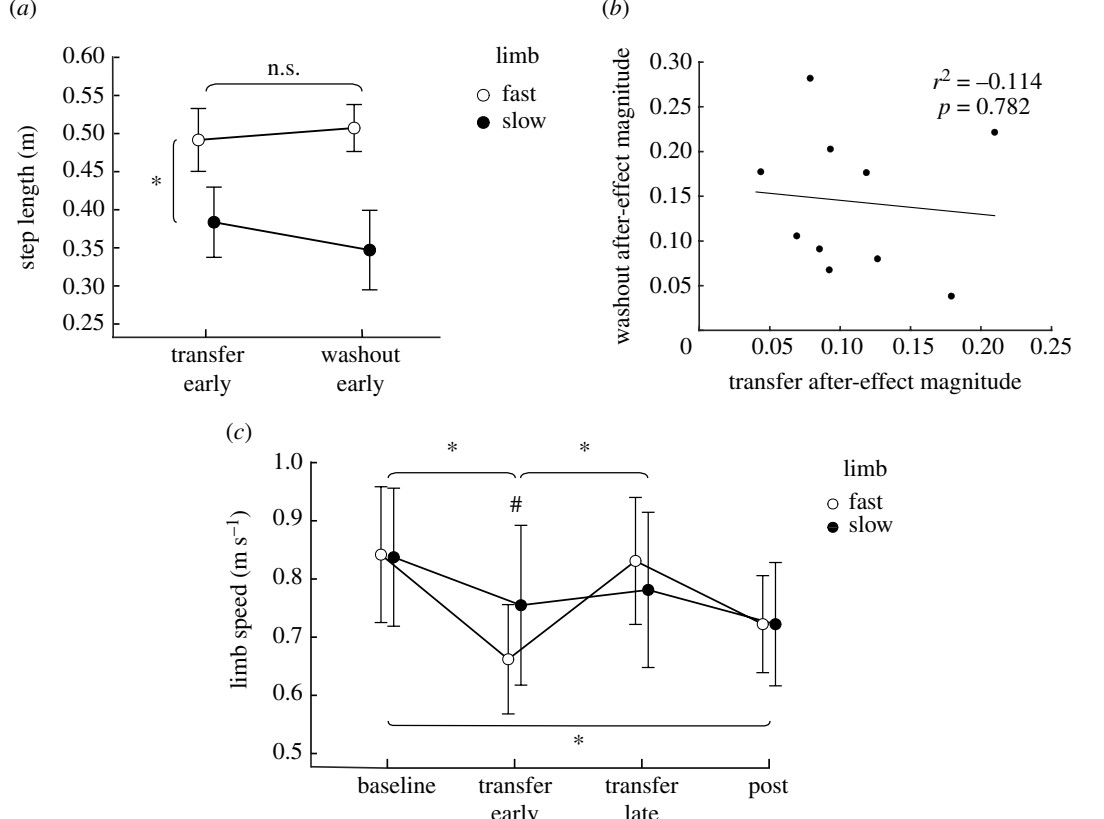

**Figure 6.** (*a*) Group average (±s.d.) step length for the fast limb (open circles) and the slow limb (closed circles) during the early transfer and early washout periods. Star indicates significant differences between pooled fast and slow limbs. (*b*) Linear regression to assess interaction between transfer after-effect and washout after-effect magnitude. Data points are individual subject averages from *early transfer-N* and *early washout-M*. (*c*) Non-motorized treadmill walking speeds across conditions. Group average (±s.d.) non-motorized treadmill belt speeds by limb across non-motorized treadmill walking conditions. Time points are during baseline non-motorized treadmill walking, *early transfer-N*, and *early post-washout-N*. *Significant difference between specified epochs and #significant difference between limbs.

slow *baseline-M* ($t = 9.1$, $p < 0.001$, figure 7*d*). By *late Adaptation-M*, double support difference was reduced back to baseline levels ($t = 0.32$, $p = 1.0$).

Figure 7*b* shows an after-effect of double support difference which was significantly different from *baseline-N* ($t = -8.0$, $p < 0.001$) and returned to baseline symmetry by *late Transfer-N* ($t = 0.07$, $p = 1.0$, figure 7*d*). Figure 7*c* shows the after-effect of double support difference during *early Washout-M* which was significantly different from slow baseline-M ($t = -8.6$, $p < 0.001$), and decreased back to baseline values by *late Washout-M* ($t = -0.19$, $p = 1.0$, figure 7*d*). See electronic supplementary material for derivation of analytical double support difference and results of individual components.

## 4. Discussion

In this study, we aimed to measure the after-effects of a split-belt locomotor adaptation in a context that allowed the limbs to move freely and express locomotor control without constraints of walking speed. In partial agreement with our hypotheses, we found robust after-effects during the non-motorized split-belt treadmill transfer condition. To our surprise, after-effects were associated with temporal but not spatial control of walking in the transfer condition, while after-effects during the washout period were associated with spatial but not temporal control. Our findings support previous studies on human locomotor adaptation, where spatial and temporal aspects of movement control are independently dissociated in healthy walking [4,5,8], as well as in impaired participants [6,19]. This is the first study to show that the adapted spatial and temporal control of walking can be independently washed out during non-motorized and motorized split-belt treadmill walking.

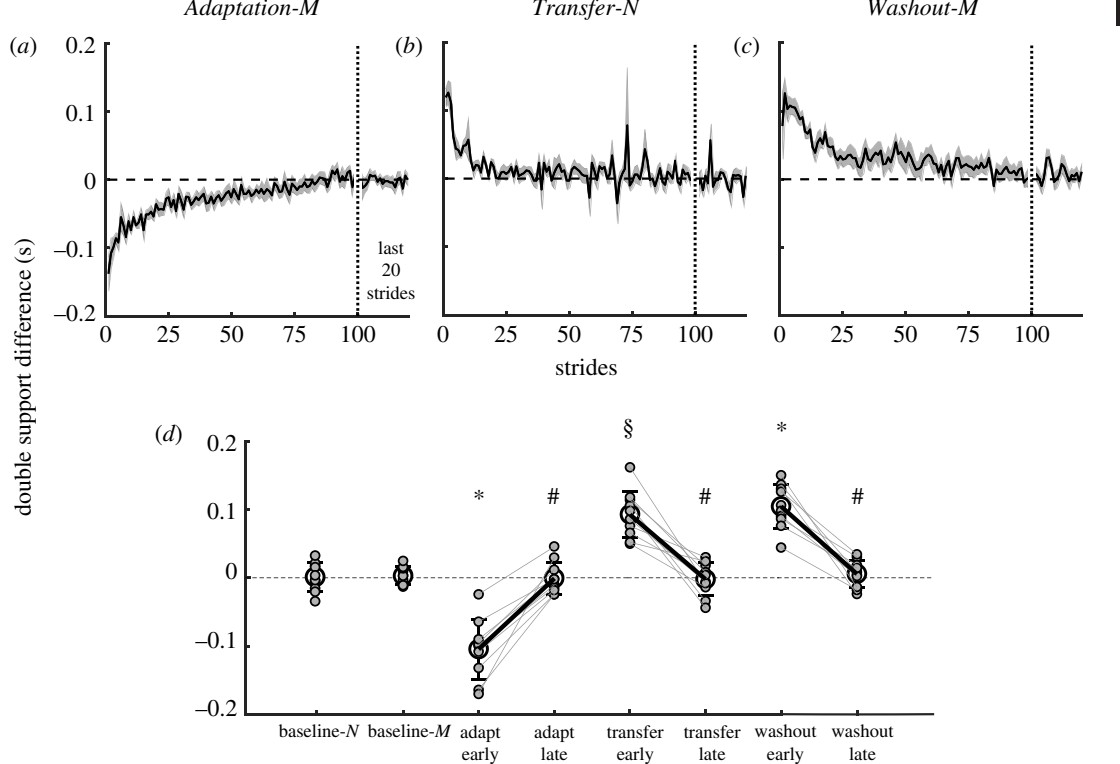

**Figure 7.** Double support difference. (a–c) Stride-by-stride data for the first 100 and final 20 strides across *Adaptation-M* (a), *Transfer-N* (b) and *Washout-M* (c). Shaded area represents ±s.e. (d) Group mean (large open circles, ±s.d.) and individual subjects data (small filled circles) across conditions. Thick black lines connect group means while thin grey lines connect individual subjects. *Significant difference between early epoch and baseline-M, §significant difference between early transfer and baseline-N and #significant different between early and late epoch.

## 4.1. Transfer of temporal but not spatial adaptation to non-motorized split-belt walking

We found a significant after-effect in both step length difference and double support difference during transfer to non-motorized split-belt walking. Notably, the after-effect during non-motorized split-belt walking was associated primarily with a change in the temporal component of step length difference, and a small but non-significant change in the velocity component. This differs from the after-effects observed during tied motorized treadmill walking at a fixed speed, where changes in both spatial and temporal control of walking has been observed [5,8]. The current findings could indicate that spatial but not temporal control is sensitive to a change in context, or that transfer of asymmetric foot placements (spatial control) was restricted due to the small non-motorized treadmill size. In either case, the temporal component was free to express the adapted asymmetry, independent of the spatial component, during the transfer condition. These results are consistent with previous findings that spatial and temporal control of walking gait are dissociable [4–6,8,20].

A recent study demonstrated that manipulation of spatial control during split-belt adaptation had no effect on temporal after-effects, but manipulation of temporal adaptation influenced the spatial after-effects [8]. If temporal asymmetries drive spatial ones, we should have seen spatial component asymmetries during transfer to non-motorized split-belt walking, but this was not the case. We believe this could be due to the unconstrained belt speeds on the non-motorized treadmill, which introduces a third degree of freedom. As mentioned above, the small treadmill space may have unintentionally blocked the spatial after-effect, and in effect pushed the asymmetry after-effect into the third degree of freedom, the limb velocity component. In support of this, our data showed that the self-selected limb speeds demonstrated after-effects during early transfer with a slower speed on the fast limb and faster speed on the slow limb. This finding is consistent with previous research which shows the perception of limb speed adapts to split-belt treadmill walking and that the fast limb during adaptation is perceived to move slower and the slow limb to move faster following adaptation, a process which may be mediated through adaptation of force-sensitive afferents [21,22]. The current

results suggest that the step length difference measure reflects a flexible combination of spatial, temporal and limb speed control, where each component can be blocked without limiting the expression of after-effects via the other components (e.g. spatial and temporal components are expressed while velocity component is blocked during motorized walking, spatial component is blocked while temporal and velocity components are expressed during non-motorized walking).

## 4.2. Persistent after-effects during motorized treadmill walking

Despite recovering baseline walking symmetry after 5 min of non-motorized split-belt walking, we observed robust after-effects in subsequent motorized tied-belt walking. We suggest three potential mechanisms contributing to persistent after-effects. If the spatial and temporal parameters were uncoupled, this would suggest independent access to the control of spatial and temporal motor outputs and independent washout of each component. In line with this idea, there were no significant changes in foot placement differences (spatial component) during transfer as step length difference after-effects were a result of changes in the temporal and velocity components. This suggests that by restricting spatial control during transfer, due to the limited deck size of the non-motorized treadmill, temporal control was independently washed out, leaving the spatial motor output to be de-adapted during the washout condition.

The apparent uncoupling of spatial and temporal after-effects could also be a re-organization of spatial, temporal and velocity asymmetries resulting from an additional degree of freedom, limb speed, when walking on the non-motorized treadmill. Close inspection of the individual components during late transfer (figure 4c,d) suggest a trade-off in spatial, temporal and velocity components, which sum to equal step lengths. By late transfer, belt speeds were asymmetric in the same direction as during adaptation, while the spatial component had a positive asymmetry (figure 4d). A positive spatial component would suggest that participants partially regain their adapted state from the motorized split-belt treadmill to trade-off symmetric step times and lengths with asymmetries in foot placement and step velocity. Trading off spatial asymmetry for temporal symmetry may be a more economical control strategy to achieve walking on the non-motorized treadmill. To support this idea, a recent study indicated that participants' self-selected temporal asymmetry during split-belt walking was energetically optimal, whereas their self-selected foot placement differences were not [20]. This would suggest that participants adapted a newly learned motor pattern to a novel condition in a way that was more efficient while meeting the demands of the novel environment, all while saving their newly adapted motor pattern for washout during the subsequent condition.

Additionally, partial washout or persistent after-effects during transfer and washout could be related to the independent after-effects associated with different walking speeds [12]. The group-averaged walking speeds during transfer ($0.91 \text{ m s}^{-1}$) were greater than that during washout ($0.67 \text{ m s}^{-1}$). Previous research has indicated that washout at the fast walking speed results in after-effects at the slow walking speed being roughly 38% of the after-effect when only washing out at the slow speed [12]. As such, washout of non-motorized treadmill walking after-effects at intermediate speed might have caused the after-effect to be partially washed out, but we found no association between the magnitude of transfer and washout after-effects. Furthermore, we found no difference in after-effect magnitudes between transfer and washout of either step length difference or double support difference, suggesting that persistent after-effects were not likely to be a function of different walking speeds.

## 4.3. Effects of inclined walking

The non-motorized treadmill is distinct from the motorized treadmill due to the *approximately 13° incline*. Because the incline is different between transfer and washout conditions, an important question is whether the incline led to artificially large after-effects which would mask partial washout or exaggerate after-effects during non-motorized treadmill walking. Increased propulsive demands of an 8.5° incline led to quicker adaptation and also led to larger after-effects relative to flat or declined walking [23,24], where the larger after-effects during tied-belts incline walking were driven by decreasing slow limb step lengths from decline to incline walking [24]. Here, we showed no difference in slow or fast step lengths between early transfer and early washout. If the after-effect during early transfer observed in the current study were driven by an excessively small slow limb step length due to the incline, we probably would have seen smaller slow limb step length during transfer compared with washout conditions. However, this was not the case. The incline of the non-motorized split-belt treadmill (*approx. 13°*) is also below the incline threshold where there is a distinct switch in intralimb

kinematics of the hip and ankle, and the timing of a group of muscles critical for forward progression during gait (rectus femoris, gluteus maximus, vastus lateralis and lateral hamstring) abruptly switches from level to incline walking [25]. We believe this suggests that the incline of the non-motorized treadmill was not a factor driving the robust after-effect during transfer.

### 4.4. Limitations

Since the hip markers were not captured during non-motorized split-belt walking, foot placement during the transfer period (estimated from de-meaned ankle data) could have been influenced by the fore–aft movement of the whole body on the treadmill. We note that while walking on the non-motorized treadmill forward and backward movement was significantly limited due to the small treadmill size and the necessity to hold the handrails to avoid sliding down the deck. Thus, we believe any small fore–aft movement on the treadmill would have a minimal effect on the outcome of the study; however, we cannot rule out this possibility.

## 5. Conclusion

We demonstrated that temporal (but not spatial) control contributed to the transfer of step length and double support adaptation from motorized to non-motorized split-belt walking. This finding is important in that it suggests independent access to neural circuitries that control temporal versus spatial aspects of gait. This is likely to translate to the clinic, where future research may address the feasibility of targeting either spatial or temporal dimensions of impaired gait separately. Further research is needed to investigate how non-motorized split-belt treadmills can be used in clinical populations, and whether training on this device would lead to improved over-ground walking.

Ethics. Informed consent was obtained by D.L.G. before the study, in accordance with the protocol that conformed to the standards of the Declaration of Helsinki (except for registration in a database) and was approved by the local Institutional Review Board at the University of Massachusetts Amherst (protocol 2015-2646).
Data accessibility. Supporting data are provided as electronic supplementary material.
Authors' contributions. D.L.G. carried out the data collections, data analysis, participated in study design and drafted the manuscript; J.T.C. participated in study design, advised the statistical analysis and critically revised the manuscript; F.C.S. advised on the building of the non-motorized treadmill and critically revised the manuscript. All authors gave final approval for publication and agree to be held accountable for the work performed therein.
Competing interests. We declare no competing interests.
Funding. This work is supported by the National Science Foundation (NSF) grant no. 1264752 to F.C.S., NSF grant no. 2001222 to J.T.C. and the Initiative for Maximizing Student Development (UMass Amherst) to D.L.G.
Acknowledgements. We thank Erin Carey, Jeffrey Florek, Nicholas Lake, Subhash Vallala and Eric Murray for designing and building the non-motorized split-belt treadmill.

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
