## [Peer Review File · Royal Society Open Science]

Review History

RSOS-200937.R0 (Original submission)

Review form: Reviewer 1

Is the manuscript scientifically sound in its present form?

Yes

Are the interpretations and conclusions justified by the results?

No

Is the language acceptable?

Yes

Do you have any ethical concerns with this paper?

No

Have you any concerns about statistical analyses in this paper?

Yes

Recommendation?

Major revision is needed (please make suggestions in comments)

Comments to the Author(s)

Title: Contributions of spatial and temporal control of step length symmetry in the transfer of locomotor adaptation from a motorized to a non-motorized split-belt treadmill

General review: in this manuscript, the authors explored transfer of split-belt walking to a non-motorized split-belt treadmill and retention in a motorized tied-belt treadmill to assess which specific aspects of adaptation (spatial or temporal) were transferred or retained. This is a very interesting manuscript using a novel experimental approach to study split-belt walking and the methods are scientifically sound. However, there are a number of changes that will improve the quality of the manuscript.

General comments:

The manuscript is written for a reader who is an expert in the field of locomotor adaptation, particularly split-belt walking experiments. A lot of experimental conditions and terms are not explicitly defined, and some terms are used interchangeably without previously defining them as equal.

There is not doubt that the experimental approach here could have implications in the field of stroke locomotion. However, this could be presented as future directions and not necessarily in the introduction as motivation to the study as not participants post-stroke were tested.

Specifically, in lines 52-61, the topic of the paragraph should be limited transfer of split belt training which is applicable to any type of population.

At the moment, it seems like the authors are developing two ideas in parallel: the feasibility of the split-belt non-motorized treadmill and transfer. The authors need to better show that they are implementing their novel non-motorized treadmill TO assess transfer AND that transfer has both a temporal and spatial domain.

The authors need to present additional rationale for their experimental hypothesis.

The authors use step length difference and step length asymmetry interchangeably but it is not clear if they refer to the same quantities. Please be consistent with terminology.

The authors conclude that non-motorized treadmill training can be used to target spatial and temporal deficits. I do not think that this can be concluded from the present experiment given that the training took place on the split-belt treadmill. This conclusion could only be reached if the experimental approach used the non-motorized treadmill for training exclusively.

I suggest comparing the results obtained here to those seen in Gonzalez-Rubio M, Velasquez NF and Torres-Oviedo G (2019) Explicit Control of Step Timing During Split-Belt Walking Reveals Interdependent Recalibration of Movements in Space and Time. *Front. Hum. Neurosci.* 13:207. doi: 10.3389/fnhum.2019.00207

The use of a self paced non-motorized treadmill is new and very interesting. Please provide information regarding speed during transfer for each leg and how it might be associated with speed during split-belt condition. This could provide some interesting information.

Detailed comments

Abstract: the abstract needs to be written for someone who is approaching this manuscript for the first time. For example, step length asymmetry, temporal and spatial control are presented in the first sentence without defining any of these concepts.

Line 26: describe experimental conditions in chronological order.

Line 26-27: replace "the ... split-belt" with "a ... split-belt"

Line 28: note that the abstract mentions step timing but this is not presented in the introduction and results.

Line 31: please provide additional information in the form of experimental results to support the finding that "transfer... were primarily driven by changes in temporal motor outputs". Also, note

that the word driven implies causality but the results only show that the temporal component was different across conditions.

The same comment applies to line 32 regarding spatial retention.

Line 33: the post-transfer condition has not been described yet.

Line 47: define positive asymmetries

Line 48: define de-adaptation

Line 49: Replace after effect with after-effects

Line 49: define post-adaptation

Line 52: the topic of this paragraph should be retention and transfer, not stroke gait, as the former is the scope of the paper

Line 56: add reference to three month retention results

Line 62: it is not clear what "these" refers to

Line 63: replace determine with set

Line 67-68: the non-motorized treadmill characteristics seem out of place in this paragraph

Line 70-72: I don't think this study can tell us about the use of non-motorized split-belt treadmills in the clinic since this study used a motorized split-belt as training.

Line 73: for the first goal, the abstract introduced transfer of step time but this is not presented here.

Line 77: make sure to differentiate between step time and the time contribution to step length asymmetry. Also, please provide the rationale for the hypothesis.

Line 78: this hypothesis is not clear. Are the authors trying to say that non motorized treadmill walking completely washes out adaptation? and if so, why is this important?

Line 100: add "to" before each

Line 111: medium speed condition not defined yet.

Line 116: how was the speed measured in the non-motorized treadmill? It would be interesting to know the speed variability for each leg on the non-motorized treadmill.

Line 126: should the condition post-adaptation be named post-washout instead?

Line 134: how were the contributions to asymmetry measured in the non-motorized treadmill without greater trochanter markers to estimate CoM?

Line 143: can a summary of the double support time findings be included in the main manuscript?

Line 160: were data normally distributed for the ANOVA? Also note that the authors present regression results and interactions but these are not described in the methods.

Also, please explicitly mention that time here is the categories defined above (and not a continuous variable).

Are the authors running separate t-tests instead of post-hoc analyses with the appropriate correction for multiple comparisons? Please clarify

Line 168: please provide summary statistics for each condition in addition to p-values

Line 176, 178, 181: what test is the p-value for?

Line 186: what does a negative after-effect refer to when SLA is positive?

Line 219: this analysis is not in the methods. What is the rationale?

Line 239: what is spatial control asymmetry?

Please summarize figure captions and verify that all information in the captions is also in the main manuscript.

Why is there a slow ramp up in step length difference during adaptation? was the speed difference introduced gradually (fig 3), or was the treadmill acceleration low? Note that this might affect the averages for Early adaptation

Review form: Reviewer 2

Is the manuscript scientifically sound in its present form?

Yes

Are the interpretations and conclusions justified by the results?

Yes

Is the language acceptable?

Yes

Do you have any ethical concerns with this paper?

No

Have you any concerns about statistical analyses in this paper?

No

Recommendation?

Major revision is needed (please make suggestions in comments)

Comments to the Author(s)

In this study, the authors built a passive, non-motorized split-belt treadmill and tested how well locomotor learning transferred from a motorized split-belt treadmill to a non-motorized split-belt treadmill. Specifically, they explored the transfer of spatial and temporal components of walking between the two conditions (and then also to normal treadmill walking afterward). They observed that learning (specifically in the temporal domain) transferred from motorized split-belt walking to non-motorized split-belt walking. This paper will be of interest to the readership of RSOS. My comments and suggestions are included below.

ABSTRACT

No comments.

INTRODUCTION

Second paragraph: the logic of the latter half of this paragraph is not entirely clear to me. How does the constraint of asymmetric belt speeds during double support limit transfer? And how would a non-motorized treadmill address this issue? It seems to me that any mechanism of split-belt walking – whether motorized or non-motorized – will have different belt speeds. I understand that this does not have to be the case in the non-motorized condition, but if the belts move at the same speed, this is not really split-belt walking and we would not expect any overground aftereffect from this type of training. Put simply: I do not understand the justification for thinking that a non-motorized split-belt treadmill may improve transfer to overground walking. This seems to be implied in the introduction.

I am also having trouble following the logic of the third paragraph. How will improving our understanding of generalization from motorized to non-motorized split-belt walking lead to implementation of non-motorized split-belt walking as a therapeutic tool? One would think that understanding how non-motorized split-belt walking generalizes to overground walking would be a more promising way to investigate this idea.

My impression from reading the entire manuscript is that the non-motorized split-belt treadmill provides an interesting way of assessing the aftereffects from split-belt treadmill walking in a

more “pure” way than is typically done by setting the belts to tied speeds using a motorized treadmill. In this way, the non-motorized treadmill is used as a measurement tool for quantifying adaptation driven by the motorized treadmill. I suggest that the authors consider framing their introduction more along these lines instead of focusing on the potential therapeutic value of the non-motorized treadmill. This is not to say that the non-motorized treadmill will not have therapeutic value, but rather that I do not think this particular study is designed to investigate the potential therapeutic value of the device.

METHODS

When calculating the alpha value (foot position), it is generally considered important to have a body-centric reference frame. For example, knowing how far the foot was placed ahead of the hip. If only measuring foot position in global space, this metric can be affected by forward/backward movement on the treadmill (e.g., it can appear that the foot was placed far ahead of the body when, in reality, the whole body moved forward). I recommend addressing this in the limitations section of the manuscript given that only ankle data were collected during the non-motorized trials.

It seems that paired t-tests would be more appropriate post-hoc tests for a repeated measures ANOVA. There should also be some correction for multiple comparisons given that multiple time points are being compared against the baseline value.

RESULTS

Line 187 – I think that “late Adaptation” should read “late Transfer”.

DISCUSSION

I suggest that the authors add discussion about the incline of the non-motorized treadmill. This is not discussed in the current draft of the manuscript, but recent work has shown that the treadmill incline/decline can have significant effects on learning (Sombric et al). Given that the non-motorized treadmill is inclined rather steeply, I suggest that this issue requires attention.

Decision letter (RSOS-200937.R0)

Dear Dr Choi:

Manuscript ID RSOS-200937 entitled "Contributions of spatial and temporal control of step length symmetry in the transfer of locomotor adaptation from a motorized to a non-motorized spli" which you submitted to Royal Society Open Science, has been reviewed. The comments from reviewers are included at the bottom of this letter.

In view of the criticisms of the reviewers, the manuscript has been rejected in its current form. However, a new manuscript may be submitted which takes into consideration these comments.

Please note that resubmitting your manuscript does not guarantee eventual acceptance, and that your resubmission will be subject to peer review before a decision is made.

Your resubmitted manuscript should be submitted by 18-Jan-2021. If you are unable to submit by this date please contact the Editorial Office.

on behalf of Dr Manoj Srinivasan (Associate Editor) and Kevin Padian (Subject Editor)
openscience@royalsociety.org

Subject Editor Comments to Author (Professor Kevin Padian):

Comments to the Author:

Thanks for your submission. The reviewers and AE like a lot of it, but there seem to be some methodological concerns, as well as questions about whether the interpretations are supported by the experiments. Although both reviewers recommended "major revision," these are usually due within three weeks and so a "reject/resub" decision allows you more leeway to consider revisions. Best success in reworking, and I hope the comments will be useful.

Reviewers' Comments to Author:

Reviewer: 1

Comments to the Author(s)

Title: Contributions of spatial and temporal control of step length symmetry in the transfer of locomotor adaptation from a motorized to a non-motorized split-belt treadmill

General review: in this manuscript, the authors explored transfer of split-belt walking to a non-motorized split-belt treadmill and retention in a motorized tied-belt treadmill to assess which specific aspects of adaptation (spatial or temporal) were transferred or retained. This is a very interesting manuscript using a novel experimental approach to study split-belt walking and the methods are scientifically sound. However, there are a number of changes that will improve the quality of the manuscript.

General comments:

The manuscript is written for a reader who is an expert in the field of locomotor adaptation, particularly split-belt walking experiments. A lot of experimental conditions and terms are not explicitly defined, and some terms are used interchangeably without previously defining them as equal.

There is not doubt that the experimental approach here could have implications in the field of stroke locomotion. However, this could be presented as future directions and not necessarily in

the introduction as motivation to the study as not participants post-stroke were tested. Specifically, in lines 52-61, the topic of the paragraph should be limited transfer of split belt training which is applicable to any type of population.

At the moment, it seems like the authors are developing two ideas in parallel: the feasibility of the split-belt non-motorized treadmill and transfer. The authors need to better show that they are implementing their novel non-motorized treadmill TO assess transfer AND that transfer has both a temporal and spatial domain.

The authors need to present additional rationale for their experimental hypothesis.

The authors use step length difference and step length asymmetry interchangeably but it is not clear if they refer to the same quantities. Please be consistent with terminology.

The authors conclude that non-motorized treadmill training can be used to target spatial and temporal deficits. I do not think that this can be concluded from the present experiment given that the training took place on the split-belt treadmill. This conclusion could only be reached if the experimental approach used the non-motorized treadmill for training exclusively.

I suggest comparing the results obtained here to those seen in Gonzalez-Rubio M, Velasquez NF and Torres-Oviedo G (2019) Explicit Control of Step Timing During Split-Belt Walking Reveals Interdependent Recalibration of Movements in Space and Time. *Front. Hum. Neurosci.* 13:207. doi: 10.3389/fnhum.2019.00207

The use of a self paced non-motorized treadmill is new and very interesting. Please provide information regarding speed during transfer for each leg and how it might be associated with speed during split-belt condition. This could provide some interesting information.

Detailed comments

Abstract: the abstract needs to be written for someone who is approaching this manuscript for the first time. For example, step length asymmetry, temporal and spatial control are presented in the first sentence without defining any of these concepts.

Line 26: describe experimental conditions in chronological order.

Line 26-27: replace "the ... split-belt" with "a ... split-belt"

Line 28: note that the abstract mentions step timing but this is not presented in the introduction and results.

Line 31: please provide additional information in the form of experimental results to support the finding that "transfer... were primarily driven by changes in temporal motor outputs". Also, note that the word driven implies causality but the results only show that the temporal component was different across conditions.

The same comment applies to line 32 regarding spatial retention.

Line 33: the post-transfer condition has not been described yet.

Line 47: define positive asymmetries

Line 48: define de-adaptation

Line 49: Replace after effect with after-effects

Line 49: define post-adaptation

Line 52: the topic of this paragraph should be retention and transfer, not stroke gait, as the former is the scope of the paper

Line 56: add reference to three month retention results

Line 62: it is not clear what "these" refers to

Line 63: replace determine with set

Line 67-68: the non-motorized treadmill characteristics seem out of place in this paragraph

Line 70-72: I don't think this study can tell us about the use of non-motorized split-belt treadmills in the clinic since this study used a motorized split-belt as training.

Line 73: for the first goal, the abstract introduced transfer of step time but this is not presented here.

Line 77: make sure to differentiate between step time and the time contribution to step length asymmetry. Also, please provide the rationale for the hypothesis.

Line 78: this hypothesis is not clear. Are the authors trying to say that non motorized treadmill walking completely washes out adaptation? and if so, why is this important?

Line 100: add "to" before each

Line 111: medium speed condition not defined yet.

Line 116: how was the speed measured in the non-motorized treadmill? It would be interesting to know the speed variability for each leg on the non-motorized treadmill.

Line 126: should the condition post-adaptation be named post-washout instead?

Line 134: how were the contributions to asymmetry measured in the non-motorized treadmill without greater trochanter markers to estimate CoM?

Line 143: can a summary of the double support time findings be included in the main manuscript?

Line 160: were data normally distributed for the ANOVA? Also note that the authors present regression results and interactions but these are not described in the methods.

Also, please explicitly mention that time here is the categories defined above (and not a continuous variable).

Are the authors running separate t-tests instead of post-hoc analyses with the appropriate correction for multiple comparisons? Please clarify

Line 168: please provide summary statistics for each condition in addition to p-values

Line 176, 178, 181: what test is the p-value for?

Line 186: what does a negative after-effect refer to when SLA is positive?

Line 219: this analysis is not in the methods. What is the rationale?

Line 239: what is spatial control asymmetry?

Please summarize figure captions and verify that all information in the captions is also in the main manuscript.

Why is there a slow ramp up in step length difference during adaptation? was the speed difference introduced gradually (fig 3), or was the treadmill acceleration low? Note that this might affect the averages for Early adaptation

Reviewer: 2

Comments to the Author(s)

In this study, the authors built a passive, non-motorized split-belt treadmill and tested how well locomotor learning transferred from a motorized split-belt treadmill to a non-motorized split-belt treadmill. Specifically, they explored the transfer of spatial and temporal components of walking between the two conditions (and then also to normal treadmill walking afterward). They observed that learning (specifically in the temporal domain) transferred from motorized split-belt walking to non-motorized split-belt walking. This paper will be of interest to the readership of RSOS. My comments and suggestions are included below.

ABSTRACT

No comments.

INTRODUCTION

Second paragraph: the logic of the latter half of this paragraph is not entirely clear to me. How does the constraint of asymmetric belt speeds during double support limit transfer? And how would a non-motorized treadmill address this issue? It seems to me that any mechanism of split-belt walking – whether motorized or non-motorized – will have different belt speeds. I understand that this does not have to be the case in the non-motorized condition, but if the belts move at the same speed, this is not really split-belt walking and we would not expect any overground aftereffect from this type of training. Put simply: I do not understand the justification

for thinking that a non-motorized split-belt treadmill may improve transfer to overground walking. This seems to be implied in the introduction.

I am also having trouble following the logic of the third paragraph. How will improving our understanding of generalization from motorized to non-motorized split-belt walking lead to implementation of non-motorized split-belt walking as a therapeutic tool? One would think that understanding how non-motorized split-belt walking generalizes to overground walking would be a more promising way to investigate this idea.

My impression from reading the entire manuscript is that the non-motorized split-belt treadmill provides an interesting way of assessing the aftereffects from split-belt treadmill walking in a more “pure” way than is typically done by setting the belts to tied speeds using a motorized treadmill. In this way, the non-motorized treadmill is used as a measurement tool for quantifying adaptation driven by the motorized treadmill. I suggest that the authors consider framing their introduction more along these lines instead of focusing on the potential therapeutic value of the non-motorized treadmill. This is not to say that the non-motorized treadmill will not have therapeutic value, but rather that I do not think this particular study is designed to investigate the potential therapeutic value of the device.

METHODS

When calculating the alpha value (foot position), it is generally considered important to have a body-centric reference frame. For example, knowing how far the foot was placed ahead of the hip. If only measuring foot position in global space, this metric can be affected by forward/backward movement on the treadmill (e.g., it can appear that the foot was placed far ahead of the body when, in reality, the whole body moved forward). I recommend addressing this in the limitations section of the manuscript given that only ankle data were collected during the non-motorized trials.

It seems that paired t-tests would be more appropriate post-hoc tests for a repeated measures ANOVA. There should also be some correction for multiple comparisons given that multiple time points are being compared against the baseline value.

RESULTS

Line 187 – I think that “late Adaptation” should read “late Transfer”.

DISCUSSION

I suggest that the authors add discussion about the incline of the non-motorized treadmill. This is not discussed in the current draft of the manuscript, but recent work has shown that the treadmill incline/decline can have significant effects on learning (Sombric et al). Given that the non-motorized treadmill is inclined rather steeply, I suggest that this issue requires attention.

Author's Response to Decision Letter for (RSOS-200937.R0)

See Appendix A.

RSOS-202084.R0

Review form: Reviewer 1

Is the manuscript scientifically sound in its present form?

Yes

Are the interpretations and conclusions justified by the results?

Yes

Is the language acceptable?

Yes

Do you have any ethical concerns with this paper?

No

Have you any concerns about statistical analyses in this paper?

No

Recommendation?

Major revision is needed (please make suggestions in comments)

Comments to the Author(s)

Thank you to the authors for their thorough work to address reviewer's comments. The new perspective that the authors have given to the manuscript is very interesting.

A personal preference regarding style. The authors have a tendency to first use terms and then define those terms either in parentheses or in a following sentence, or using i.e., so I found myself having to go back and forth during the reading. It would help improve clarity and flow if the authors defined terms before using them to make the flow of information linear, or if they explicitly used the definitions instead of expressing the idea once and then explaining what the idea means. See for example the abstract, line 84, line 86, etc

I found myself constantly having to go back to figure 1 to remind myself whether each of the conditions took place on the motorized or non-motorized treadmill. I think including the treadmill type in the condition name would greatly improve readability (for example baseline-M and baseline-NM or washout-M, post-washout-NM).

Detailed comments

Abstract

The abstract is a bit jargon-y and requires first reading the paper to understand the abstract. Remember that the abstract is the first (and sometimes only) part of the paper the reader encounters. The abstract can be written in a way that a person with a basic understanding of the field can follow.

All the clarifications in parentheses can probably be omitted (see main comment).

The authors can probably generalize the experimental conditions in the abstract to say something along the lines of "participants adapted to a split-belt treadmill and then we measured aftereffects during non-motorized treadmill walking and tied belt walking". Particularly, given that the authors only present results for two conditions in the abstract, it seems like there is a mismatch between methods and results. Also note that the post-transfer condition is never defined in the abstract.

Line 45: should the first sentence be: locomotion is highly adaptable? In its current way it is unclear how the topic sentence regarding neural control of locomotion relates to the following sentence and the overall paragraph.

Line 60-61: I think the authors need to expand on this idea of which specific variables can affect transfer and summarize the results in the cited manuscripts.

Line 81: replace alternatively with additionally as this hypothesis is not in contrast with the previous one.

Line 86 – 90: what is the rationale for this hypothesis? Is there literature that might support the idea that if participants can move their leg at their preferred speed would lead to symmetric step lengths?

Line 137 – 140: it is not clear why the speed clarifications are provided and how these two sentences are different. Was pilot testing done as part of this study?

Line 148: this sentence should go before the section where the authors present the split-belt condition.

167: does this refer to limb velocity or to treadmill velocity? Note that the limb velocity equals the treadmill velocity only during stance phase.

Line 199: does this refer to the ipsilateral hip markers or the average position in the X direction of the hip markers? Note that given pelvic rotation, this would generate different position values

Line 201 – 205: this needs to be validated using data for treadmill walking and presented in supplementary materials. The authors can calculate limb position during treadmill walking using the hip markers and the average of the ankle markers. Results could differ between motorized and non-motorized walking because of the difference in calculations.

Line 202: does de-meanded refer to the fact that the average in the x direction of the two markers was subtracted? Or the average of the timeseries of the ipsilateral marker.

Line 214: the authors need to acknowledge the slow change in step length difference when the belts are split. Also, as mentioned in the response to reviewers, this slow change occurs over 10 strides, meaning that the use of the first 5 strides to calculate values during early adaptation might not measure the true magnitude of the initial perturbation. Based on this, the largest asymmetry does not occur during early adaptation.

Line 215 – 216: the authors mention that their statistics are testing differences in DV between epochs. What is the scientific question that each difference between epochs answers?

Line 225: it is unclear to me how this analysis will dissociate the effects of incline. I think the only way to test for the effects of incline is to walk on the same treadmill with or without an incline.

Line 248: the authors present changes in spatial, temp and vel components relative to baseline. note however that the baseline components have not been presented.

Figure 3 (applies to all figures) are a bit confusing given that one panel shows SE and the other shows SD. I suggest using SD for all figures to show the true spread of the data. for figure 3D, what are the error bars?

Figure 4: typo in x-ticks "trasnfer"

Are the baselines in figure 3, 4, 5 the same data? Please specify whether baselines are for motorized and non motorized treadmills both in the figures and text. Also, how do the spatiotemporal characteristics for tied belt walking at 1m/s compare to the non-motorized treadmill walking at 1m/s?

Line 271: this effect size calculation was not included in the statistical analysis section of the methods. what is the goal of this analysis and why was it only done for step length difference during transfer?

I suggest adding a title with the specific condition for all panels in figures 7

Please see the link here on how to structure the discussion.

<https://www.biosciencewriters.com/How-to-Write-a-Strong-Discussion-in-Scientific-Manuscripts.aspx> The first paragraph should present the big picture and second paragraph should present the critical findings. The specific sections presented in the manuscript should come after the second paragraph. I found the discussion a bit hard to follow

Line 367: what does a modest change in context refer to? This term is subjective. One might argue that a TM that the participant has to actively propel is not a modest change in context.

Line 404: how was spatial control restricted during transfer? Please clarify what restricted refers to.

Line 407: does next refer to the second hypothesized mechanism introduced in the previous paragraph?

Line 413 – 418: these statements are not clear.

please add legends to figures to indicate what the different colors mean

Review form: Reviewer 2

Is the manuscript scientifically sound in its present form?

Yes

Are the interpretations and conclusions justified by the results?

Yes

Is the language acceptable?

Yes

Do you have any ethical concerns with this paper?

No

Have you any concerns about statistical analyses in this paper?

No

Recommendation?

Accept as is

Comments to the Author(s)

The authors have done a nice job responding to the original reviewer comments. I have no additional suggestions and thank the authors for sharing their interesting work; this is a cool study!

Decision letter (RSOS-202084.R0)

Dear Dr Choi

The Editors assigned to your paper RSOS-202084 "Contributions of spatial and temporal control in the transfer of locomotor adaptation from motorized to non-motorized split-belt walking" have now received comments from reviewers and would like you to revise the paper in accordance with the reviewer comments and any comments from the Editors. Please note this decision does not guarantee eventual acceptance.

Please submit your revised manuscript and required files (see below) no later than 21 days from today's (ie 11-Dec-2020) date. Note: the ScholarOne system will 'lock' if submission of the revision is attempted 21 or more days after the deadline. If you do not think you will be able to meet this deadline please contact the editorial office immediately.

on behalf of Dr Manoj Srinivasan (Associate Editor) and Kevin Padian (Subject Editor)
openscience@royalsociety.org

Associate Editor Comments to Author (Dr Manoj Srinivasan):

Thank you for your revisions. Both reviewers were pleased with how you addressed their concerns. One of the reviewers has provided a few further comments/suggestions and asked for clarifications of a few technical claims and interpretations. We look forward to a revision that addresses these, which should be addressable with small edits to the manuscript. This reviewer also had further stylistic comments which I encourage the authors to consider when possible to improve readability.

Reviewer comments to Author:

Reviewer: 2

Comments to the Author(s)

The authors have done a nice job responding to the original reviewer comments. I have no additional suggestions and thank the authors for sharing their interesting work; this is a cool study!

Reviewer: 1
 Comments to the Author(s)

Thank you to the authors for their thorough work to address reviewer's comments. The new perspective that the authors have given to the manuscript is very interesting.

A personal preference regarding style. The authors have a tendency to first use terms and then define those terms either in parentheses or in a following sentence, or using i.e., so I found myself having to go back and forth during the reading. It would help improve clarity and flow if the authors defined terms before using them to make the flow of information linear, or if they explicitly used the definitions instead of expressing the idea once and then explaining what the idea means. See for example the abstract, line 84, line 86, etc

I found myself constantly having to go back to figure 1 to remind myself whether each of the conditions took place on the motorized or non-motorized treadmill. I think including the treadmill type in the condition name would greatly improve readability (for example baseline-M and baseline-NM or washout-M, post-washout-NM).

Detailed comments

Abstract

The abstract is a bit jargon-y and requires first reading the paper to understand the abstract. Remember that the abstract is the first (and sometimes only) part of the paper the reader encounters. The abstract can be written in a way that a person with a basic understanding of the field can follow.

All the clarifications in parentheses can probably be omitted (see main comment).

The authors can probably generalize the experimental conditions in the abstract to say something along the lines of "participants adapted to a split-belt treadmill and then we measured aftereffects during non-motorized treadmill walking and tied belt walking". Particularly, given that the authors only present results for two conditions in the abstract, it seems like there is a mismatch between methods and results. Also note that the post-transfer condition is never defined in the abstract.

Line 45: should the first sentence be: locomotion is highly adaptable? In its current way it is unclear how the topic sentence regarding neural control of locomotion relates to the following sentence and the overall paragraph.

Line 60-61: I think the authors need to expand on this idea of which specific variables can affect transfer and summarize the results in the cited manuscripts.

Line 81: replace alternatively with additionally as this hypothesis is not in contrast with the previous one.

Line 86 - 90: what is the rationale for this hypothesis? Is there literature that might support the idea that if participants can move their leg at their preferred speed would lead to symmetric step lengths?

Line 137 - 140: it is not clear why the speed clarifications are provided and how these two sentences are different. Was pilot testing done as part of this study?

Line 148: this sentence should go before the section where the authors present the split-belt condition.

167: does this refer to limb velocity or to treadmill velocity? Note that the limb velocity equals the treadmill velocity only during stance phase.

Line 199: does this refer to the ipsilateral hip markers or the average position in the X direction of the hip markers? Note that given pelvic rotation, this would generate different position values

Line 201 - 205: this needs to be validated using data for treadmill walking and presented in supplementary materials. The authors can calculate limb position during treadmill walking using the hip markers and the average of the ankle markers. Results could differ between motorized and non-motorized walking because of the difference in calculations.

Line 202: does de-meanded refer to the fact that the average in the x direction of the two markers was subtracted? Or the average of the timeseries of the ipsilateral marker.

Line 214: the authors need to acknowledge the slow change in step length difference when the belts are split. Also, as mentioned in the response to reviewers, this slow change occurs over 10 strides, meaning that the use of the first 5 strides to calculate values during early adaptation might not measure the true magnitude of the initial perturbation. Based on this, the largest asymmetry does not occur during early adaptation.

Line 215 – 216: the authors mention that their statistics are testing differences in DV between epochs. What is the scientific question that each difference between epochs answers?

Line 225: it is unclear to me how this analysis will dissociate the effects of incline. I think the only way to test for the effects of incline is to walk on the same treadmill with or without an incline.

Line 248: the authors present changes in spatial, temp and vel components relative to baseline. note however that the baseline components have not been presented.

Figure 3 (applies to all figures) are a bit confusing given that one panel shows SE and the other shows SD. I suggest using SD for all figures to show the true spread of the data. for figure 3D, what are the error bars?

Figure 4: typo in x-ticks "trasfer"

Are the baselines in figure 3, 4, 5 the same data? Please specify whether baselines are for motorized and non motorized treadmills both in the figures and text. Also, how do the spatiotemporal characteristics for tied belt walking at 1m/s compare to the non-motorized treadmill walking at 1m/s?

Line 271: this effect size calculation was not included in the statistical analysis section of the methods. what is the goal of this analysis and why was it only done for step length difference during transfer?

I suggest adding a title with the specific condition for all panels in figures 7

Please see the link here on how to structure the discussion.

<https://www.biosciencewriters.com/How-to-Write-a-Strong-Discussion-in-Scientific-Manuscripts.aspx> The first paragraph should present the big picture and second paragraph should present the critical findings. The specific sections presented in the manuscript should come after the second paragraph. I found the discussion a bit hard to follow

Line 367: what does a modest change in context refer to? This term is subjective. One might argue that a TM that the participant has to actively propel is not a modest change in context.

Line 404: how was spatial control restricted during transfer? Please clarify what restricted refers to.

Line 407: does next refer to the second hypothesized mechanism introduced in the previous paragraph?

Line 413 – 418: these statements are not clear.

please add legends to figures to indicate what the different colors mean

===PREPARING YOUR MANUSCRIPT===

Please ensure that you include an acknowledgements' section before your reference list/bibliography. This should acknowledge anyone who assisted with your work, but does not

qualify as an author per the guidelines at <https://royalsociety.org/journals/ethics-policies/openness/>.

===PREPARING YOUR REVISION IN SCHOLARONE===

- Ensure that your data access statement meets the requirements at <https://royalsociety.org/journals/authors/author-guidelines/#data>. You should ensure that you cite the dataset in your reference list. If you have deposited data etc in the Dryad repository, please include both the 'For publication' link and 'For review' link at this stage.
- If you are requesting an article processing charge waiver, you must select the relevant waiver option (if requesting a discretionary waiver, the form should have been uploaded at Step 3 'File upload' above).
- If you have uploaded ESM files, please ensure you follow the guidance at <https://royalsociety.org/journals/authors/author-guidelines/#supplementary-material> to include a suitable title and informative caption. An example of appropriate titling and captioning may be found at https://figshare.com/articles/Table_S2_from_Is_there_a_trade-off_between_peak_performance_and_performance_breadth_across_temperatures_for_aerobic_sc_ope_in_teleost_fishes_/3843624.

Author's Response to Decision Letter for (RSOS-202084.R0)

See Appendix B.

RSOS-202084.R1 (Revision)

Review form: Reviewer 1

Is the manuscript scientifically sound in its present form?

Yes

Are the interpretations and conclusions justified by the results?

Yes

Is the language acceptable?

Yes

Do you have any ethical concerns with this paper?

No

Have you any concerns about statistical analyses in this paper?

No

Recommendation?

Accept as is

Comments to the Author(s)

I would like to thank the authors for their efforts to address all reviewer comments. All my comments have been carefully addressed

Decision letter (RSOS-202084.R1)

Dear Dr Choi,

It is a pleasure to accept your manuscript entitled "Contributions of spatial and temporal control in the transfer of locomotor adaptation from motorized to non-motorized split-belt walking" in its current form for publication in Royal Society Open Science.

on behalf of Dr Manoj Srinivasan (Associate Editor) and Kevin Padian (Subject Editor)
openscience@royalsociety.org

Associate Editor Comments to Author (Dr Manoj Srinivasan):

Associate Editor: 1

Comments to the Author:

Thank you to the authors for carefully addressing all the reviewer and editorial comments.

Reviewer comments to Author:

Reviewer: 1

Comments to the Author(s)

I would like to thank the authors for their efforts to address all reviewer comments. All my comments have been carefully addressed

Appendix A

Subject Editor Comments to Author (Professor Kevin Padian):

Comments to the Author:

Thanks for your submission. The reviewers and AE like a lot of it, but there seem to be some methodological concerns, as well as questions about whether the interpretations are supported by the experiments. Although both reviewers recommended "major revision," these are usually due within three weeks and so a "reject/resub" decision allows you more leeway to consider revisions. Best success in reworking, and I hope the comments will be useful.

We would like to thank the editor and reviewers for their time and feedback for this manuscript. The comments are very helpful, and we have revised the paper to address all of the points that were raised. Major changes to the manuscript include:

- a. Revised the Introduction to focus on the novel experimental approach of using a non-motorized split-belt treadmill to assess locomotor adaptation transfer from a basic science point-of-view
- b. Added new text and figures to provide information regarding individual step length during early transfer and washout (Fig. 6a) and the speed of each leg during non-motorized treadmill walking conditions (Fig. 6c).
- c. Added double support findings (Fig. 7) to the Results section
- d. Clarified statistical analyses in both the Methods and Results sections
- e. Revised the interpretation and conclusion in the Discussion section in light of recent studies by Gonzalez-Rubio *et al.* 2019 and Sombric *et al.* 2019.

We believe the manuscript has been significantly improved and hope that our changes reflect your thoughtful critiques adequately. Please find below our responses to each comment (numbered and restated in **bold**). In some cases, excerpts from the revised text is included in the responses (in *italics*). Changes to the manuscript are highlighted in **red** text.

Reviewer #1 Comments to Author:

General review: in this manuscript, the authors explored transfer of split-belt walking to a non-motorized split-belt treadmill and retention in a motorized tied-belt treadmill to assess which specific aspects of adaptation (spatial or temporal) were transferred or retained. This is a very interesting manuscript using a novel experimental approach to study split-belt walking and the methods are scientifically sound. However, there are a number of changes that will improve the quality of the manuscript.

General comments:

1. The manuscript is written for a reader who is an expert in the field of locomotor adaptation, particularly split-belt walking experiments. A lot of experimental conditions and terms are not explicitly defined, and some terms are used interchangeably without previously defining them as equal.

The conditions and terms are now explicitly defined to enhance readability.

2. **There is not doubt that the experimental approach here could have implications in the field of stroke locomotion. However, this could be presented as future directions and not necessarily in the introduction as motivation to the study as not participants post-stroke were tested. Specifically, in lines 52-61, the topic of the paragraph should be limited transfer of split belt training which is applicable to any type of population.**

Thank you for this comment. Reviewer #2 share the same sentiment. We have revised the Introduction to focus on the basic science aspects of this study and the novel use of a non-motorized treadmill to assess transfer of adaptations from the motorized split-belt treadmill.

3. **At the moment, it seems like the authors are developing two ideas in parallel: the feasibility of the split-belt non-motorized treadmill and transfer. The authors need to better show that they are implementing their novel non-motorized treadmill TO assess transfer AND that transfer has both a temporal and spatial domain.**

We made changes throughout the manuscript to highlight the novelty of the non-motorized treadmill for assessing transfer of adaptations from the motorized split-belt treadmill. The Introduction has been updated to elaborate on the assessment of adaptation transfer in a less constraining walking environment using the non-motorized treadmill. The Discussion has been revised to better explain the temporal and spatial aspects of the transfer.

4. **The authors need to present additional rationale for their experimental hypothesis.**

We have revised this paragraph as follows on p.4:

“In this study, we examined the transfer of adaptation from motorized split-belt treadmill walking at a 2:1 speed ratio to non-motorized split-belt walking at self-selected speeds on each side. We predicted at least three potential outcomes. (1) We hypothesized that, in the absence of asymmetrical changes in self-selected leg speeds during non-motorized split-belt walking, the transfer of adapted foot placement (spatial control) and step timing (temporal control) from motorized to non-motorized split-belt walking would result in asymmetrical step lengths—i.e., typical after-effects from motorized split-belt adaptation would be observed during transfer to non-motorized split-belt walking (Gonzalez-Rubio et al., 2019). (2) Alternatively, we hypothesized that self-selected speed would increase on the fast leg and decrease on the slow leg. Here ‘fast’ and ‘slow’ legs refer to the leg on the fast and slow belts, respectively, during motorized split-belt adaptation. Asymmetrical changes in leg speed that complement the adapted foot placement and step time would result in symmetrical step lengths during non-motorized split-belt walking—i.e., the adapted spatiotemporal walking pattern during motorized split-belt walking at 2:1 speed ratio would persist during transfer to non-motorized split-belt walking. (3) In addition, the adapted foot placement and step timing contributions to step length difference may show only partial (incomplete) transfer to non-motorized split-belt walking. Partial transfer and washout of after-effects would result in residual after-effects during subsequent tied-belt motorized split-belt walking (Vasudevan and Bastian, 2010).”

5. **The authors use step length difference and step length asymmetry interchangeably but it is not clear if they refer to the same quantities. Please be consistent with terminology.**

We have made changes throughout the manuscript to be clearer when referring to step length difference versus the step length symmetry.

- 6. The authors conclude that non-motorized treadmill training can be used to target spatial and temporal deficits. I do not think that this can be concluded from the present experiment given that the training took place on the split-belt treadmill. This conclusion could only be reached if the experimental approach used the non-motorized treadmill for training exclusively.**

I suggest comparing the results obtained here to those seen in Gonzalez-Rubio M, Velasquez NF and Torres-Oviedo G (2019) Explicit Control of Step Timing During Split-Belt Walking Reveals Interdependent Recalibration of Movements in Space and Time. *Front. Hum. Neurosci.* 13:207. doi: 10.3389/fnhum.2019.00207

Thank you for the feedback. We now address the results of the current study with those of Gonzalez-Rubio and Torres-Oviedo 2019 in the discussion, second paragraph, starting at line 373.

"A recent study demonstrated that manipulation of spatial control during split-belt adaptation had no effect on temporal after-effects, but manipulation of temporal adaptation influenced the spatial after-effects (Gonzalez-Rubio et al., 2019). If temporal asymmetries drive spatial ones, we should have seen spatial component asymmetries during transfer to non-motorized split-belt walking, but this was not the case. We believe this could be due to the unconstrained belt speeds on the non-motorized treadmill, which introduces a third degree of freedom. As mentioned above, the small treadmill space may have unintentionally blocked the spatial after-effect, and in effect pushed the asymmetry after-effect into the third degree of freedom—i.e., limb velocity component. In support of this, our data showed that the self-selected limb speeds demonstrated after-effects during early transfer with a slower speed on the fast limb and faster speed on the slow limb. This finding is consistent with previous research which shows the perception of limb speed adapts to split-belt treadmill walking and that the fast limb during adaptation is perceived to move slower and the slow limb to move faster following adaptation, a process which may be mediated through adaptation of force sensitive afferents (Jensen et al., 1998; Vazquez et al., 2015). The current results suggest that step length difference reflects a flexible combination of spatial, temporal and limb speed control, where each component can be blocked without limiting the expression of after-effects via the other components (e.g., spatial and temporal components are expressed while velocity component is blocked during motorized walking, spatial component is blocked while temporal and velocity components are expressed during non-motorized walking)"

- 7. The use of a self paced non-motorized treadmill is new and very interesting. Please provide information regarding speed during transfer for each leg and how it might be associated with speed during split-belt condition. This could provide some interesting information.**

Thank you for this comment. As suggested, we have added information regarding the speed of each leg during non-motorized split-belt walking. Please see the new Figure 6c. in the Results section. We noted a trend with a slower speed on the 'fast limb' and fast speed on the 'slow limb' during transfer.

Detailed comments

- 8. Abstract: the abstract needs to be written for someone who is approaching this manuscript for the first time. For example, step length asymmetry, temporal and spatial control are presented in the first sentence without defining any of these concepts.**

The abstract has been revised with more accessible language for a wider audience. Thank you for helping to improve the writing quality.

- 9. Line 26: describe experimental conditions in chronological order.**

Experimental conditions are now listed in chronological order, lines 30-33.

- 10. Line 26-27: replace “the ... split-belt” with “a ... split-belt”**
“the” replaced with “a”

- 11. Line 28: note that the abstract mentions step timing but this is not presented in the introduction and results.**

We have clarified the text in the abstract to indicate that “temporal control” is a construct of step timing. This construct is further elucidated in the methods section, lines 185-193, where we discuss that step length difference is the sum of a spatial, temporal, and velocity components—of which the temporal component comprises the difference in slow and fast step times as a function of average foot speed. In the results section, the temporal component is indicated as the step time component or step time contribution and is referenced as such throughout the results section.

- 12. Line 31: please provide additional information in the form of experimental results to support the finding that “transfer... were primarily driven by changes in temporal motor outputs”. Also, note that the word driven implies causality but the results only show that the temporal component was different across conditions.**

We modified the language from “driven” to “associated with”.

- 13. The same comment applies to line 32 regarding spatial retention.**

We made the same changes here as in the comment above.

- 14. Line 33: the post-transfer condition has not been described yet.**

Language about the post-transfer (i.e. washout) condition have been added in line 33.

- 15. Line 47: define positive asymmetries**

This was removed from the revised Introduction. Positive asymmetries are now defined in the Methods section.

- 16. Line 48: define de-adaptation**

This section has been revised.

- 17. Line 49: Replace after effect with after-effects**

This section has been revised.

- 18. Line 49: define post-adaptation**

This section has been revised.

- 19. Line 52: the topic of this paragraph should be retention and transfer, not stroke gait, as the former is the scope of the paper**
Thank you for the suggestion. We have made major revisions to the introduction to improve the scope and concise nature of the paper.
- 20. Line 56: add reference to three month retention results**
This was removed from the revised Introduction.
- 21. Line 62: it is not clear what “these” refers to**
This section has been revised.
- 22. Line 63: replace determine with set**
This section has been revised.
- 23. Line 67-68: the non-motorized treadmill characteristics seem out of place in this paragraph**
This was removed from the revised Introduction.
- 24. Line 70-72: I don’t think this study can tell us about the use of non-motorized split-belt treadmills in the clinic since this study used a motorized split-belt as training.**
This sentence has also been removed in the revised introduction to improve the overall message of using the non-motorized treadmill to better understand transfer from motorized split-belt treadmills.
- 25. Line 73: for the first goal, the abstract introduced transfer of step time but this is not presented here.**
This section has been revised.
- 26. Line 77: make sure to differentiate between step time and the time contribution to step length asymmetry. Also, please provide the rationale for the hypothesis.**
Thank you for pointing this out. The distinction between step time and the temporal contribution to step length difference are now defined (p.8, lines 190-192). We have also clarified the hypothesis: please see our response to #6 above.
- 27. Line 78: this hypothesis is not clear. Are the authors trying to say that non motorized treadmill walking completely washes out adaptation? and if so, why is this important?**
Complete washout would indicate adaptation of shared neural circuits for motorized and non-motorized walking, while lack of washout would indicate adaptation of separate neural circuits from those used in non-motorized treadmill walking. We have improved the clarity of our hypothesis in the third paragraph of the introduction and have provided a specific reference which suggests that this may be the case (i.e. Vasudevan and Bastian 2010). Please also see response to comment #4 above.
- 28. Line 100: add “to” before each**
“to” added before “each” in line 116.
- 29. Line 111: medium speed condition not defined yet.**
Added “(1.0 m/s)” after “speed” in line 136-137 to clarify that the medium speed is 1.0 meters per second.

- 30. Line 116: how was the speed measured in the non-motorized treadmill? It would be interesting to know the speed variability for each leg on the non-motorized treadmill.**
Limb speed during walking on the non-motorized treadmill was defined as the instantaneous speed of the ankle marker at the time of contralateral heel strike (line 167-169). We have also added measures of limb speed changes during non-motorized treadmill walking (figure 6c). Please also see our response to #7 above.
- 31. Line 126: should the condition post-adaptation be named post-washout instead?**
Thank you for pointing this out, “post-adaptation” changed to “post-washout”, line 147.
- 32. Line 134: how were the contributions to asymmetry measured in the non-motorized treadmill without greater trochanter markers to estimate CoM?**
We have clarified in the methods section that during non-motorized treadmill walking, the ankle data was de-meant to artificially create a “body-centered” frame from which to calculate the alpha parameter (p.9, lines 201-203). We have also included a discussion of this limitation, starting line 454.
- 33. Line 143: can a summary of the double support time findings be included in the main manuscript?**
Thank you for suggesting the addition of the double support findings. This addition of double support in the Results (p.14) will improve the understanding of the project.
- 34. Line 160: were data normally distributed for the ANOVA? Also note that the authors present regression results and interactions but these are not described in the methods.**
The results from the ANOVA indicate that the data were normally distributed as Mauchly’s test of sphericity was not violated, indicated in lines 237-239

Thank you for pointing out that we neglected to include the statistics for the linear regression for the interaction of transfer and washout after-effects. We have updated the statistical analysis section to include these methods in lines 222-225.
- 35. Also, please explicitly mention that time here is the categories defined above (and not a continuous variable).**
“time” has been reframed as “epoch” throughout the results section to improve clarity.
- 36. Are the authors running separate t-tests instead of post-hoc analyses with the appropriate correction for multiple comparisons? Please clarify**
Thank you for pointing this out. We re-ran our post-hoc analysis with Bonferroni corrections to account for multiple comparisons. We have updated the statistical analysis section to reflect this change. Performing the post-hoc tests with Bonferroni corrections changed two of our previous findings:
1) The step length difference during the late transfer period was significantly different from baseline but is no longer different. See results, lines 265-267.
2) The velocity component during early and late transfer was significantly different from baseline but is no longer different when using the updated post-hoc tests. See results, lines 269-272. Reference to significant change in the velocity component has also been removed from the abstract and an updated discussion of the implications of this have been made in the discussion section, lines 365-366.
- 37. Line 168: please provide summary statistics for each condition in addition to p-values**

Summary statistics added for each result throughout the results section.

38. Line 176, 178, 181: what test is the p-value for?

Results have been updated to include summary statistics, and more clearly indicate the test comparison being made.

39. Line 186: what does a negative after-effect refer to when SLA is positive?

This line was meant to indicate that the after-effect was in the opposite direction of the initial asymmetry. “a negative after-effect” changed to “an after-effect” in line 264 to improve clarity.

40. Line 219: this analysis is not in the methods. What is the rationale?

Thank you for pointing out the missing information. We have updated the methods section to include this analysis. Rationale is also included in the last paragraph of the new introduction. See also, response to comments #4 and #34 above.

41. Line 239: what is spatial control asymmetry?

This section has been revised.

42. Please summarize figure captions and verify that all information in the captions is also in the main manuscript.

Excess text has been removed from figure captions and language made more concise., Relevant information has also been moved to the main text of the methods section of the manuscript.

43. Why is there a slow ramp up in step length difference during adaptation? was the speed difference introduced gradually (fig 3), or was the treadmill acceleration low? Note that this might affect the averages for Early adaptation

The slow ramp in step length difference which occurs during the first ~10 strides is a result of the early foot placement differences during that time (see figure 3.c, red spatial component). We are not sure why that behavior emerged from this group of participants but seems to be driven by about half of the subjects who display larger than normal variability of foot placement difference during that period.

To clarify about the methodology used during motorized treadmill walking, we included a sentence to indicate that the introduction to the split-belt environment started from zero to indicated speed at 1.0 m/s² in lines 136-137.

Review #2 Comments to the Author(s)

In this study, the authors built a passive, non-motorized split-belt treadmill and tested how well locomotor learning transferred from a motorized split-belt treadmill to a non-motorized split-belt treadmill. Specifically, they explored the transfer of spatial and temporal components of walking between the two conditions (and then also to normal treadmill walking afterward). They observed that learning (specifically in the temporal domain) transferred from motorized split-belt walking to non-motorized split-belt walking. This paper will be of interest to the readership of RSOS. My comments and suggestions are included below.

ABSTRACT

No comments.

INTRODUCTION

1. **Second paragraph: the logic of the latter half of this paragraph is not entirely clear to me. How does the constraint of asymmetric belt speeds during double support limit transfer? And how would a non-motorized treadmill address this issue? It seems to me that any mechanism of split-belt walking – whether motorized or non-motorized – will have different belt speeds. I understand that this does not have to be the case in the non-motorized condition, but if the belts move at the same speed, this is not really split-belt walking and we would not expect any overground aftereffect from this type of training. Put simply: I do not understand the justification for thinking that a non-motorized split-belt treadmill may improve transfer to overground walking. This seems to be implied in the introduction.**

Thank you for pointing this out. The introduction has been revised to allow a more clear framing of the novelty and use of the non-motorized treadmill to assess a more natural way to understand motorized split-belt treadmill adaptations. The second paragraph has been revised as follows (lines 63-75):

“During non-motorized split-belt treadmill walking, the speed of each belt/limb is user controlled. This novel device thus allows us to examine the adaptation and storage of spatial and temporal motor outputs during walking in the absence of leg-specific speed constraints. Previous research indicated that after-effects from split-belt walking adaptation are modulated by speed constraints, with the largest after-effects occurring while both legs walk at the slow speed (Vasudevan and Bastian, 2010). Using the non-motorized split-belt treadmill, we can assess whether self-selected walking speeds are also modulated after motorized split-belt walking adaptation, and how self-selected limb speeds influence after-effect magnitude.”

2. **I am also having trouble following the logic of the third paragraph. How will improving our understanding of generalization from motorized to non-motorized split-belt walking lead to implementation of non-motorized split-belt walking as a therapeutic tool? One would think that understanding how non-motorized split-belt walking generalizes to overground walking would be a more promising way to investigate this idea.**

Thank you for pointing this out. We have made significant modifications to the introduction to be more clear and the novelty and use of the non-motorized treadmill to assess the locomotor adaptation to split-belt walking in a more pure way as you mention in #3 below.

3. **My impression from reading the entire manuscript is that the non-motorized split-belt treadmill provides an interesting way of assessing the aftereffects from split-belt treadmill walking in a more “pure” way than is typically done by setting the belts to tied speeds using a motorized treadmill. In this way, the non-motorized treadmill is used as a measurement tool for quantifying adaptation driven by the motorized treadmill. I suggest that the authors consider framing their introduction more along these lines instead of focusing on the potential therapeutic value of the non-motorized treadmill. This is not to say that the non-motorized treadmill will not have**

therapeutic value, but rather that I do not think this particular study is designed to investigate the potential therapeutic value of the device.

Thank you for these suggestions. Reviewer #1 share the same sentiments. Again, we have made significant modifications to the introduction to re-frame the subject of the manuscript and the use of the non-motorized treadmill as a way to assess adaptations of motorize split-belt treadmill walking.

METHODS

- 4. When calculating the alpha value (foot position), it is generally considered important to have a body-centric reference frame. For example, knowing how far the foot was placed ahead of the hip. If only measuring foot position in global space, this metric can be affected by forward/backward movement on the treadmill (e.g., it can appear that the foot was placed far ahead of the body when, in reality, the whole body moved forward). I recommend addressing this in the limitations section of the manuscript given that only ankle data were collected during the non-motorized trials.**

We have clarified in the methods section that during non-motorized treadmill walking, the ankle data was de-meant to artificially create a “body-centered” frame from which to calculate the alpha parameter (p.9, lines 201-203).

We have also included a discussion of this in the limitations section as follows (lines 454-461):

“Since the hip markers were not captured during non-motorized split-belt walking, foot placement during the transfer period (estimated from de-meant ankle data) could have been influenced by the fore-aft movement of the whole body on the treadmill. We note that while walking on the non-motorized treadmill forward and backward movement on the was significantly limited due to the small treadmill size and the necessity to hold the handrails to avoid sliding down the deck. Thus, we believe any small fore-aft movement on the treadmill would have a minimal effect on the outcome of the study, however we cannot rule out this possibility.”

- 5. It seems that paired t-tests would be more appropriate post-hoc tests for a repeated measures ANOVA. There should also be some correction for multiple comparisons given that multiple time points are being compared against the baseline value.**

Thank you for pointing this out along with Reviewer 1. We re-ran our post-hoc analysis with Bonferroni corrections to account for multiple comparisons. We have updated the statistical analysis section in lines 216-225 to reflect this change. Performing the post-hoc tests with Bonferroni corrections changed two of our previous findings:

1) The step length difference during the late transfer period was significantly different from baseline but is no longer different. See results, lines 265-267.

2) The velocity component during early and late transfer was significantly different from baseline but is no longer different when using the updated post-hoc tests. See results, lines 270-273. Reference to significant change in the velocity component has also been removed from the abstract and an updated discussion of the implications of this have been made in the discussion section, lines 365-366.

RESULTS

6. Line 187 – I think that “late Adaptation” should read “late Transfer”.

“late Adaptation” changed to “late Transfer” in line 265.

DISCUSSION

7. I suggest that the authors add discussion about the incline of the non-motorized treadmill. This is not discussed in the current draft of the manuscript, but recent work has shown that the treadmill incline/decline can have significant effects on learning (Sombric et al). Given that the non-motorized treadmill is inclined rather steeply, I suggest that this issue requires attention.

Thank you for the suggestion. We have added a new section to the discussion on the implications of the incline in lines 435-451 addressing this. Further, we have added a statistical test and figure (figure 6A) comparing individual step lengths from early transfer and early washout for both the fast and slow limbs to support our discussion:

“The non-motorized treadmill is distinct from the motorized treadmill due to the ~13° incline. Because the incline is different between transfer and washout conditions, an important question is whether the incline led to artificially large after-effects which would mask partial washout or exaggerate after-effects during non-motorized treadmill walking. Increased propulsive demands of an 8.5 degree incline led to quicker adaptation and also led to larger after-effects relative to flat or declined walking (Sombric and Torres-Oviedo, 2020; Sombric et al., 2019), where the larger after-effects during tied-belts incline walking were driven by decreasing slow limb step lengths from decline to incline walking (Sombric et al., 2019). Here, we showed no difference in slow or fast step lengths between early transfer and early washout. If the after-effect during early transfer observed in the current study were driven by an excessively small slow limb step length due to the incline, we likely would have seen smaller slow limb step length during transfer compared to washout conditions. However, this was not the case. The incline of the non-motorized split-belt treadmill (~13°) is also below the incline threshold where there is a distinct switch in intralimb kinematics of the hip and ankle, and the timing of a group of muscles critical for forward progression during gait (rectus femoris, gluteus maximus, vastus lateralis, and lateral hamstring) abruptly switches from level to incline walking (Earhart and Bastian, 2000).”

Appendix B

Associate Editor Comments to Author (Dr Manoj Srinivasan):

Thank you for your revisions. Both reviewers were pleased with how you addressed their concerns. One of the reviewers has provided a few further comments/suggestions and asked for clarifications of a few technical claims and interpretations. We look forward to a revision that addresses these, which should be addressable with small edits to the manuscript. This reviewer also had further stylistic comments which I encourage the authors to consider when possible to improve readability.

We would like to thank Dr. Srinivasan and the reviewers for the additional feedback. Major changes to the second submission include:

- a. Added supporting materials comparing the two different methods used to calculate the spatial component of step length difference. See response to comment #12.
- b. Revised figures so that all error bars represent standard deviation.
- c. Added supplemental table summarizing statistical analyses.

Please find below our responses to each comment (numbered and restated in bold). In some cases, excerpts from the revised text is included in the responses (in *italics*). Changes to the manuscript are highlighted in red text.

Reviewer: 2

The authors have done a nice job responding to the original reviewer comments. I have no additional suggestions and thank the authors for sharing their interesting work; this is a cool study!

Thank you for your feedback. Your suggestions have made this manuscript better.

Reviewer: 1

Thank you to the authors for their thorough work to address reviewer's comments. The new perspective that the authors have given to the manuscript is very interesting.

A personal preference regarding style. The authors have a tendency to first use terms and then define those terms either in parentheses or in a following sentence, or using i.e., so I found myself having to go back and forth during the reading. It would help improve clarity and flow if the authors defined terms before using them to make the flow of information linear, or if they explicitly used the definitions instead of expressing the idea once and then explaining what the idea means. See for example the abstract, line 84, line 86, etc

Thank you for the suggestion. We have made changes throughout the manuscript to be more concise in language and have limited the use of the "i.e." style to improve flow of reading in the manuscript.

I found myself constantly having to go back to figure 1 to remind myself whether each of the conditions took place on the motorized or non-motorized treadmill. I think including the treadmill type in the condition name would greatly improve readability (for example baseline-M and baseline-NM or washout-M, post-washout-NM).

We have made changes throughout the methods and results sections to indicate whether a condition is on the motorized (M) or non-motorized (N) treadmills as suggested.

Detailed comments

Abstract

1. **The abstract is a bit jargon-y and requires first reading the paper to understand the abstract. Remember that the abstract is the first (and sometimes only) part of the paper the reader encounters. The abstract can be written in a way that a person with a basic understanding of the field can follow.**

The abstract has been revised to be more accessible to the average reader.

2. **All the clarifications in parentheses can probably be omitted (see main comment).**

Clarifications in parentheses has been removed where possible.

3. **The authors can probably generalize the experimental conditions in the abstract to say something along the lines of “participants adapted to a split-belt treadmill and then we measured aftereffects during non-motorized treadmill walking and tied belt walking”. Particularly, given that the authors only present results for two conditions in the abstract, it seems like there is a mismatch between methods and results. Also note that the post-transfer condition is never defined in the abstract.**

Thank you for the wording suggestion. We modified the text to say

“10 healthy participants adapted on a motorized split-belt treadmill (2:1 speed ratio) and were then assessed for after-effects during subsequent non-motorized treadmill and motorized tied-belt treadmill walking.”

Language to post-transfer has been removed and rephrased to

“Conversely, residual after-effects during motorized tied-belt walking following transfer were associated with foot placement differences.”

4. **Line 45: should the first sentence be: locomotion is highly adaptable? In its current way it is unclear how the topic sentence regarding neural control of locomotion relates to the following sentence and the overall paragraph.**

Opening sentence to the introduction starting line 45 has been rephrased to

“A healthy human nervous system can rapidly adapt how the limbs move to meet changing environmental demands.”

5. **Line 60-61: I think the authors need to expand on this idea of which specific variables can affect transfer and summarize the results in the cited manuscripts.**

We have added a paragraph to expand upon the effects of different aspects of the transfer environment to the presence of after-effects in split-belt treadmill walking. The following is an excerpt from the revised introduction:

“Changes in walking condition, such as the speed of walking, can lead to diminishing after-effects as the speed deviates from that of the slow limb during adaptation (12). Further, after-effect magnitude during transfer is known to be modulated by contextual clues such as when transferring to over-ground walking, or when vision is manipulated between training and transfer contexts (11,14). The removal of vision during training and transfer contexts maximizes transfer in healthy adults, but when only removed during

training or transfer conditions limits the transfer of spatial (i.e., step length) or temporal (i.e. phase shift) parameters respectively (11).”

6. Line 81: replace alternatively with additionally as this hypothesis is not in contrast with the previous one.

We have kept the wording as was originally written because hypothesis one is in opposition to hypothesis two. Hypothesis one states that with symmetric limb speeds, persistent foot placement and step timing after-effects will lead to asymmetric step lengths. This potential outcome resembles that of slow washout walking where the limbs move at the same speed during stance, but because foot placement and step timing asymmetries persist, step length asymmetry is a constrained consequence because step length is the sum of the differences in foot placements on consecutive steps with the product of step time and limb speed. In effect, we have transfer of spatio-temporal control but not “walking speed”, which results in after-effects such as what is seen during tied slow washout walking. However, in hypothesis two, we argue that a second possibility is that both limb speed and spatiotemporal control transfer, in which case step lengths between the limbs are roughly equal, same as seen during motorized split-belt walking where limb speeds, foot placements, and step times are asymmetric. This then creates a dichotomy where participants either transfer the entirety of split-belt walking control and persist with their motorized split-belt limb control which will later washout during slow motorized treadmill walking, or they transfer only the spatio-temporal control but not limb speed control which leads to a washout of their adapted pattern. With our third hypothesis, we then have i) partial transfer and full washout, ii) full transfer and no washout, or iii) partial transfer and partial washout (also see response to #7 below).

7. Line 86 – 90: what is the rationale for this hypothesis? Is there literature that might support the idea that if participants can move their leg at their preferred speed would lead to symmetric step lengths?

Thank you for this question as we realize we have not been as clear as we intend. We have rephrased our hypotheses to match our explanation from 6 above as follows:

“We predicted three potential outcomes. (i) Partial transfer and full washout. We hypothesized that, in the absence of asymmetrical changes in self-selected leg speeds during non-motorized split-belt walking, the transfer of adapted foot placement (spatial control) and step timing (temporal control) from motorized to non-motorized split-belt walking would result in asymmetrical step length after-effects typical of motorized split-belt adaptation and would lead to the washout of the adapted motor pattern (8). (ii) Full transfer and no washout. Alternatively, we hypothesized that self-selected speed would increase on the leg that was on the fast belt and decrease on the leg that was on the slow belt to match speeds during adaptation. Asymmetric changes in leg speed that complement the adapted foot placement and step time would result in symmetrical step lengths during non-motorized split-belt walking such that the adapted spatiotemporal walking pattern during motorized split-belt walking at 2:1 speed ratio would persist during transfer to non-motorized split-belt walking and not washout. (iii) Partial transfer and partial washout. In addition, the adapted foot placement and step timing contributions to step length difference may show only partial (incomplete) transfer to non-motorized split-belt walking. Partial transfer and washout of after-effects would result in residual after-effects during subsequent tied-belt motorized split-belt walking.”

8. Line 137 – 140: it is not clear why the speed clarifications are provided and how these two sentences are different. Was pilot testing done as part of this study?

In lines 149 to 151, we first state “These speeds were chosen because the average speed is 1.0 m/s, which was the average preferred walking speed on the non-motorized treadmill during pilot testing.” Here, we are indicating that the slow and fast speeds during motorized treadmill split-belt walking have an average speed of 1.0 m/s which is the average speed recorded during preferred non-motorized treadmill walking during pilot testing conducted for this study. The following sentence has been removed as it incorrectly stated that baseline non-motorized treadmill walking speed had an average of 1.0 m/s. However, figure 6, panel “C” demonstrates that the average non-motorized treadmill walking speed during baseline-N walking was close to 0.8 m/s.

9. Line 148: this sentence should go before the section where the authors present the split-belt condition.

Sentence moved to the last sentence of the Experimental Setup opening paragraph, starting at line 118.

10. 167: does this refer to limb velocity or to treadmill velocity? Note that the limb velocity equals the treadmill velocity only during stance phase.

We have rephrased this sentence to be more explicit:

“stance limb velocity during walking on the non-motorized treadmill was calculated as the instantaneous speed of the ankle marker at the time of contralateral heel strike.”

11. Line 199: does this refer to the ipsilateral hip markers or the average position in the X direction of the hip markers? Note that given pelvic rotation, this would generate different position values

Thank you for pointing out that the language here can be more clear. We have rephrased to:

“Note that for the step length analysis during motorized treadmill walking, we estimated the center of mass position by taking the average position of the two hip markers; the anterior-posterior center of mass position was then subtracted from each ankle marker to express foot position in the center of mass reference frame.”

12. Line 201 – 205: this needs to be validated using data for treadmill walking and presented in supplementary materials. The authors can calculate limb position during treadmill walking using the hip markers and the average of the ankle markers. Results could differ between motorized and non-motorized walking because of the difference in calculations.

First, we would like to mention that different epochs of walking during Adaptation, Transfer and Washout were compared to their respective baseline, so that we are comparing values calculated using the same method in each analysis. We have now included a statistical table in the supplemental materials to summarize these comparisons.

Second, we did verify and found a strong agreement between the ankle- and hip-based spatial component using data collected during motorized treadmill walking. The plot below shows the comparison using strides from baseline (slow, medium, and fast), and washout periods during motorized treadmill walking from one subject:

- A foot placement is the position of a foot at heel strike in a specified reference frame (i.e., either center of mass space or ankle space) and is thus subject to fore-aft treadmill movement (panel 1A-B). Alpha is the placement of a foot relative to the previous foot's placement and is less sensitive to fore-aft movement on the treadmill (panel 1C-D). The spatial component is calculated as the difference of two successive alpha measures (figure

R2); therefore the spatial component is a difference of foot placement differences (see Finley et al. 2015).

- In figure 1A, foot placement is calculated from the de-meaned ankle data and fore-aft treadmill movement is clear in comparison to foot placement in center of mass space (fig. 1B). Despite the fore-aft movement on the motorized treadmill, the difference between the alpha value of each limb calculated with either the ankle or hip methods are similar (fig. 1C & 1D). In panel 1E we show that the two methods have a strong agreement with an r^2 of 0.75 and slope 0.98. Using the Bland-Altman method[§], in panel 1F we show that there is no systematic offset with a zero-mean difference between measures ($p = 0.33$), there is no proportional error trend, and we show that the variability is consistent. The coefficient of reproducibility (RPC) is 0.02, which is below the sample standard deviation (0.0227).

[§]Reference: Giavarina D. Understanding Bland Altman analysis. Biochem Med (Zagreb). 2015 Jun 5;25(2):141-51.

- We also verified this for the alpha differences (i.e., the spatial parameter in our paper). In figure 2B we show that there is strong agreement between methods for slow washout walking with an $r^2 = 0.71$, zero mean difference ($p = 0.68$), and an RPC of 0.03.

- 13. Line 202: does de-meanded refer to the fact that the average in the x direction of the two markers was subtracted? Or the average of the timeseries of the ipsilateral marker.**

We have rephrased this sentence to be more clear:

“the average value of the ankle marker across a trial from the limb assigned to the slow belt during motorized treadmill walking was subtracted from each ankle markers anterior-posterior position to obtain a “body centered reference frame.”

- 14. Line 214: the authors need to acknowledge the slow change in step length difference when the belts are split. Also, as mentioned in the response to reviewers, this slow change occurs over 10 strides, meaning that the use of the first 5 strides to calculate values during early adaptation might not measure the true magnitude of the initial perturbation. Based on this, the largest asymmetry does not occur during early adaptation.**

We have made a note in the Results section, in lines 252-256, that the planned analysis of using the first 5 strides underestimates the magnitude of the initial asymmetry induced by the split-belt condition:

“Note that during the early Adaptation period, step length difference shows a gradual increase in asymmetry from stride 1 to about stride 10. Therefore, our planned analysis

using the first 5 strides during early Adaptation is an under-estimate of the maximal asymmetry achieved during split-belt treadmill walking.”

15. Line 215 – 216: the authors mention that their statistics are testing differences in DV between epochs. What is the scientific question that each difference between epochs answers?

This is to test whether a variable is different from baseline, and whether there is a change from early to late Adaptation/Transfer/Washout. Changes across baseline, early- and late- Adaptation is used to determine adaptation of walking variables during the speed perturbation. Change across baseline, early- and late- Transfer/Washout is used to determine storage and washout of after-effects from the adaptation. We have clarified this as follows:

“Separate repeated measures ANOVAs were used to test the effects of epoch on step length difference and its components. Post-hoc analyses with Bonferroni corrections were used to assess for differences between each epoch compared to the corresponding baseline (e.g., baseline-M vs. early adapt, baseline-M vs. late adapt), and between early and late epochs (e.g., early adapt vs. late adapt) when significant main effects were determined.”

16. Line 225: it is unclear to me how this analysis will dissociate the effects of incline. I think the only way to test for the effects of incline is to walk on the same treadmill with or without an incline.

We have removed the reference to using this test to assess the effects of the incline. We have rephrased the sentence to:

“We also performed a repeated measures ANOVA to test the effects of epoch (Transfer-N early, Washout-M early) and limb on individual step lengths as a comparison of previous split-belt treadmill research involving the use of an incline.”

We further discuss the implications of this comparison in the section “Effects of inclined walking” and reference the research by Sombric and others.

17. Line 248: the authors present changes in spatial, temp and vel components relative to baseline. note however that the baseline components have not been presented.

Thank you for pointing out the missing statistical analysis. Statistical comparisons of the baseline individual components to zero have been added in lines 249-251.

Additionally, we have added information regarding the Students’ t-test’s that were used to make baseline comparisons to zero in the statistical analysis section in lines 227-228.

18. Figure 3 (applies to all figures) are a bit confusing given that one panel shows SE and the other shows SD. I suggest using SD for all figures to show the true spread of the data. for figure 3D, what are the error bars?

We have re-made figures 3, 4, and 5 with all error bars on panels B and D in SD (panel D, formerly SE). We chose to keep the stride-by-stride data in panels A and C as standard error to improve the visibility of the stride-by-stride data differences between individual components. We clarified in each figure caption what shading (SE) and error bars (SD) indicate. We have also changed the figure title to indicate what data each condition is referring to.

19. Figure 4: typo in x-ticks “trasnfer”

Figure 3-5 x-ticks have been modified to remove reference to the condition since it is in the figure title.

20. Are the baselines in figure 3, 4, 5 the same data? Please specify whether baselines are for motorized and non motorized treadmills both in the figures and text. Also, how do the spatiotemporal characteristics for tied belt walking at 1m/s compare to the non-motorized treadmill walking at 1m/s?

Thank you for pointing out that the baselines were not specified. We compared Adaptation-M (fig. 3) and Washout-M (fig. 5) to the slow motorized baseline, and we compared Transfer-N (fig. 4) to the non-motorized baseline. We now label the motorized and non-motorized baselines as “baseline-M” and “baseline-N”, respectively, in each figure.

The spatiotemporal parameters were symmetric during baseline tied-belt walking at 1 m/s, as well as non-motorized treadmill walking at preferred speed baseline-N. We indicate this in the Results as follows:

“Group averaged step-length difference and its components were not significantly different from zero during baseline motorized and non-motorized treadmill walking trials (all p-values > 0.05).”

21. Line 271: this effect size calculation was not included in the statistical analysis section of the methods. what is the goal of this analysis and why was it only done for step length difference during transfer?

We have removed the reference to the effects size and instead refer the reader to the statistical table in Supplementary materials. The description of the calculation for the effects size is included after the supplementary table.

22. I suggest adding a title with the specific condition for all panels in figures 7

The condition titles are now added to each panel in figure 7. We have also added the baseline non-motorized treadmill walking data to figure 7D and changed how a significant difference is indicated between early-baseline and early-late comparisons to improve the clarity of the figure.

23. Please see the link here on how to structure the discussion.

<https://www.biosciencewriters.com/How-to-Write-a-Strong-Discussion-in-Scientific-Manuscripts.aspx> The first paragraph should present the big picture and second paragraph should present the critical findings. The specific sections presented in the manuscript should come after the second paragraph. I found the discussion a bit hard to follow

Thank you for the excellent reference for writing a discussion. We have added a paragraph to the beginning of the discussion to improve the flow and readability of our discussion section while opening our individual discussion points more clearly.

24. Line 367: what does a modest change in context refer to? This term is subjective. One might argue that a TM that the participant has to actively propel is not a modest change in context.

The word “modest” has been removed.

25. Line 404: how was spatial control restricted during transfer? Please clarify what restricted refers to.

We have rephrased the sentence to indicate that “restricted” during non-motorized treadmill walking refers to the small treadmill deck surface:

“This suggests that by restricting spatial control during Transfer, due to the limited deck size of the non-motorized treadmill, temporal control was independently washed out, leaving the spatial motor output to be deadadapted during the Washout condition.”

26. Line 407: does next refer to the second hypothesized mechanism introduced in the previous paragraph?

We removed the “First” from line 426 and the “Next” from line 443 has been changed to “Additionally” to improve the clarity of each sentence and the flow of the discussion.

27. Line 413 – 418: these statements are not clear.

We have rephrased this section to more clearly indicate that we suggest a spatial asymmetry was chosen over a temporal one (as seen during early transfer) to be more economical:

“A positive spatial component would suggest that participants partially regain their adapted state from the motorized split-belt treadmill to trade-off symmetric step times and lengths with asymmetries in foot placement and step velocity. Trading off spatial asymmetry for temporal symmetry may be a more economical control strategy to achieve walking on the non-motorized treadmill. To support this idea, a recent study indicated that participants’ self-selected temporal asymmetry during split-belt walking was energetically optimal whereas their self-selected foot placement differences were not (19).”

28. please add legends to figures to indicate what the different colors mean

Legends added to figures 3-5 to inform the reader of the colors in the figure.